# XLF acts as a flexible connector during non-homologous end joining

**Sean M Carney[1], Andrew T Moreno[1], Sadie C Piatt[1,2], Metztli Cisneros-Aguirre[3,4], Felicia Wednesday Lopezcolorado[3], Jeremy M Stark[3,4], Joseph J Loparo[1]***

[1]Department of Biological Chemistry and Molecular Pharmacology, Blavatnik Institute, Harvard Medical School, Boston, United States; [2]Harvard Graduate Program in Biophysics, Harvard Medical School, Boston, United States; [3]Department of Cancer Genetics and Epigenetics, Beckman Research Institute of the City of Hope, Duarte, United States; [4]Irell and Manella Graduate School of Biological Sciences, Beckman Research Institute of the City of Hope, Duarte, United States

**Abstract** Non-homologous end joining (NHEJ) is the predominant pathway that repairs DNA double-strand breaks in vertebrates. During NHEJ DNA ends are held together by a multi-protein synaptic complex until they are ligated. Here, we use *Xenopus laevis* egg extract to investigate the role of the intrinsically disordered C-terminal tail of the XRCC4-like factor (XLF), a critical factor in end synapsis. We demonstrate that the XLF tail along with the Ku-binding motif (KBM) at the extreme C-terminus are required for end joining. Although the underlying sequence of the tail can be varied, a minimal tail length is required for NHEJ. Single-molecule FRET experiments that observe end synapsis in real-time show that this defect is due to a failure to closely align DNA ends. Our data supports a model in which a single C-terminal tail tethers XLF to Ku, while allowing XLF to form interactions with XRCC4 that enable synaptic complex formation.

**\*For correspondence:**
joseph_loparo@hms.harvard.edu

**Competing interests:** The authors declare that no competing interests exist.

## Introduction

DNA double-strand breaks (DSBs) are a particularly toxic form of DNA damage. Within vertebrates the majority of DSBs are repaired by non-homologous end joining (NHEJ) (*Rothkamm et al., 2003*). In contrast to homologous recombination (HR), the other major DSB repair pathway, NHEJ does not use a DNA template to guide repair. Instead, a synaptic complex comprised of core and accessory NHEJ factors holds DNA ends together until they are ultimately ligated. Recognition of the DSB is carried out by the ring-shaped Ku70/Ku80 heterodimer (Ku) which rapidly binds DNA ends (*Walker et al., 2001*) and subsequently recruits downstream NHEJ factors including the DNA-dependent protein kinase catalytic subunit, DNA-PKcs (*Uematsu et al., 2007*; *Jette and Lees-Miller, 2015*), whose kinase activity is essential for DNA repair (*Jiang et al., 2015*). As DSBs arise from a wide-range of sources, DNA ends are often initially incompatible with ligation. A host of end processing enzymes, including NHEJ-associated polymerases and nucleases, act on these ends to allow for ligation by DNA ligase IV (Lig4) (*Grawunder et al., 1997*; *Stinson et al., 2020*). Ligation requires at least two additional factors: XRCC4, a scaffolding factor to which Lig4 is constitutively bound and the structurally related XRCC4-like factor (XLF) (*Grawunder et al., 1997*; *Buck et al., 2006*; *Ahnesorg et al., 2006*). Together these factors must assemble into a synaptic complex that recognizes, synapses, aligns, processes, and ligates DNA ends.

The NHEJ synaptic complex holds DNA ends together through a complicated network of inter-molecular interactions. Emerging single-molecule approaches in cell-free extracts and reconstitutions have provided new mechanistic details of how these interactions evolve during repair reactions. Using single-molecule Förster resonance energy transfer (smFRET) experiments to monitor the

distance between DNA ends in *Xenopus laevis* egg extract, we have shown that there are at least two distinct synaptic states that precede ligation (*Graham et al., 2016*). First, Ku and DNA-PKcs (but not kinase activity) are required to weakly tether DNA ends at a distance where they are protected from processing, a state we named the long-range (LR) complex (*Stinson et al., 2020*; *Graham et al., 2016*). Next, DNA-PKcs kinase activity, XRCC4, Lig4, and XLF are required to transition from the initial LR complex to a stable short-range (SR) synaptic complex in which the DNA ends are closely aligned for processing and ligation (*Stinson et al., 2020*; *Graham et al., 2016*). Importantly, the catalytic activity of Lig4 is not required to form the SR complex, demonstrating that the ligase plays a structural role in end synapsis (*Graham et al., 2016*; *Cottarel et al., 2013*). Subsequent biochemical reconstitutions of human NHEJ proteins also found evidence for these two synaptic states (*Wang et al., 2018*), suggesting that the architecture of the NHEJ synaptic complex is conserved from *Xenopus* to humans.

While XLF has been implicated in DNA end synapsis, there are conflicting models of its role in NHEJ. XLF, as well as the structurally similar XRCC4 and PAXX, exist as homodimers (*Xing et al., 2015*; *Ochi et al., 2015*; *Li et al., 2008*). Each of these paralogues consists of an N-terminal globular head domain, an extended coiled-coil stalk that mediates dimerization (*Li et al., 2008*), and a flexible C-terminal tail (*Figure 1A*; *Andres et al., 2007*). XLF and XRCC4 interact through their head domains (*Andres et al., 2012*; *Malivert et al., 2010*), and this interaction is essential for NHEJ in cells (*Roy et al., 2015*; *Roy et al., 2012*). In minimal reconstitutions of XRCC4 and XLF, this interaction can lead to the formation of extensive XLF-XRCC4 filaments (*Andres et al., 2012*; *Ropars et al., 2011*), which have been proposed to be important for repair in vivo (*Mahaney et al.,*

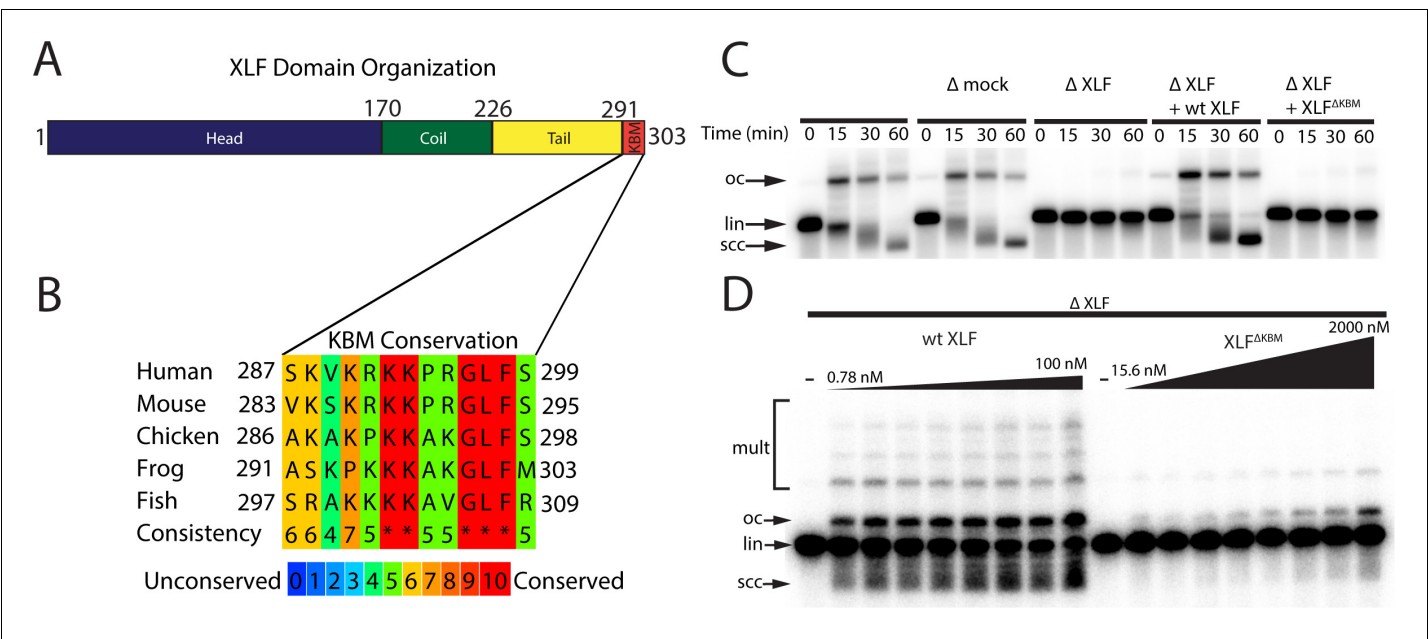

**Figure 1.** The Ku binding motif (KBM) of XLF is critical for end joining in *Xenopus* egg extract. (**A**) A schematic of the domain organization of XLF. The residue number at the boundary of each region is shown. (**B**) A protein sequence alignment of the KBM from human (*Homo sapiens* UniProt ID Q9H9Q4), mouse (*Mus musculus* UniProt ID Q3KNJ2), chicken (Gallus gallus UniProt ID F1NVP8), frog (*Xenopus laevis* see note below), and fish (Danio rerio UniProt ID Q6NV18). The *Xenopus laevis* XLF sequence and translation start site was determined previously by immunoprecipitation from extract and subsequent trypsin digestion and analysis by mass spectrometry (*Graham et al., 2016*). The alignment was performed using PRALINE multiple sequence alignment and its default settings (*Simossis and Heringa, 2005*). A colored scale shows the degree of conservation. (**C**) Ensemble time course end joining assay in either mock-depleted (immunodepletion with non-specific rabbit IgG) or XLF-depleted *Xenopus* egg extract. Recombinant wild-type (wt) XLF and XLF$^{\Delta KBM}$ were added back at 75 nM final concentration. DNA species: scc, supercoiled closed circular; lin, linear; oc, open circle. (**D**) Ensemble end point titration end joining assay in *Xenopus* egg extract. XLF-depleted extract was supplemented with recombinant protein, either wt XLF or XLF$^{\Delta KBM}$, at varying concentrations. The reactions were stopped after 20 min. DNA species: scc, supercoiled closed circular; lin, linear; oc, open circle; mult, multimer.

The online version of this article includes the following figure supplement(s) for figure 1:

**Figure supplement 1.** *Xenopus laevis* XLF is evolutionarily conserved and can be depleted from egg extract.

*2013*; *Reid et al., 2015*). However, single-molecule imaging of fluorescently labeled XLF in egg extract by our laboratory revealed that a single XLF dimer is sufficient for SR complex formation, although it must interact with XRRC4 through both of its head domains for optimal end joining (*Graham et al., 2018*). These results suggest that a single XLF dimer makes several contacts within the SR complex to mediate synapsis.

XLF contains an extended intrinsically disordered C-terminal tail with a highly conserved Ku-binding motif (KBM) at the terminus (*Figure 1A–B*; *Yano et al., 2011*; *Nemoz et al., 2018*). Deletion of the KBM ablates cellular recruitment of XLF to sites of DNA damage (*Yano et al., 2011*) and mutations within the KBM result in varying defects in cellular end joining assays and cell survival in the presence of DSB inducing agents (*Nemoz et al., 2018*; *Bhargava et al., 2018*). The contributions of the remainder of the C-terminal tail, henceforth referred to as the tail, are unknown, although it has been shown to be a target for phosphorylation by DNA-PKcs (*Yu et al., 2008*). Across vertebrate species, there is poor sequence conservation within this region, and yet the tail is invariably retained with a similar length (*Figure 1—figure supplement 1A*). Deleting the entire C-terminal region of XLF (including the KBM) still allows for interaction with XRCC4 (*Andres et al., 2007*). However, minimal reconstitutions show that the C-terminal region of XLF is necessary for an interaction with DNA in vitro (*Andres et al., 2007*; *Andres et al., 2012*) and variably stimulates (1.5–40 fold) end joining (*Wang et al., 2018*; *Andres et al., 2007*). It remains unclear if these defects arise solely from the loss of the XLF KBM or if the tail also contributes to NHEJ.

Here, we investigate the role of the XLF C-terminal tail in NHEJ synaptic complex assembly. We show that both the KBM and the tail are essential for robust end joining in both *Xenopus* egg extract and in cells. Ablation of potential phosphorylation sites and shuffling the sequence of the tail region did not alter repair efficiency, demonstrating that, independent of sequence, the length of this region is important for NHEJ. Using asymmetric mutants of XLF, we show that a single KBM within the XLF dimer is sufficient for end joining. These observations rule out models where both XLF tails are required to span the break via interactions with opposing Ku molecules. Instead, the length of the tail is required for XLF-mediated stabilization of XRCC4-Lig4 at DNA ends. We propose that the XLF tail acts to tether XLF to DSBs through its interaction with Ku while simultaneously allowing XLF to interact with XRCC4 and drive formation of the SR complex.

## Results

### The KBM of XLF is essential for end joining in *Xenopus* egg extract

To test whether the KBM of XLF is necessary for end joining, we performed a time course end joining assay in *Xenopus* egg extract. In this assay, we monitored the conversion of radiolabeled linear DNA to joined products (open circular DNA, supercoiled closed DNA, and in some cases multimers) over time (*Figure 1C*). Immunodepletion of XLF from extract (*Figure 1—figure supplement 1B*) abolished end joining (*Figure 1C*), consistent with prior results in extract and in cells (*Graham et al., 2016*; *Roy et al., 2015*). A mock immunodepletion did not alter end joining, and joining in XLF-depleted extract was rescued by adding back physiological concentrations (~75 nM; *Wühr et al., 2014*) of recombinant wild type (wt) XLF (*Figure 1C*). Notably, an XLF mutant lacking the KBM, XLF$^{\Delta KBM}$, did not rescue end joining in XLF-depleted extract, demonstrating that the KBM is necessary for end joining (*Figure 1C*). To further characterize the severity of this defect, we performed titrations of wt XLF and XLF$^{\Delta KBM}$ in XLF-depleted extract (*Figure 1D*). While wt XLF was able to robustly rescue joining at sub-nanomolar concentrations, the XLF$^{\Delta KBM}$ supported little end joining even at micromolar concentrations. Collectively, these results are consistent with the KBM playing a critical role in recruiting XLF to DSBs through its interaction with Ku (*Yano et al., 2011*; *Nemoz et al., 2018*).

### The interior region of the C-terminal tail of XLF is essential for end joining

Given the importance of the KBM, we next tested whether the tail of XLF is also required for end joining. We constructed several mutants in which the tail was truncated while retaining the KBM at the very C-terminus (*Figure 2A*). The three truncation constructs XLF$^{1-285+KBM}$, XLF$^{1-265+KBM}$, and XLF$^{1-245+KBM}$ removed 6, 26, and 46 residues respectively from the tail (*Figure 2A*). To determine if

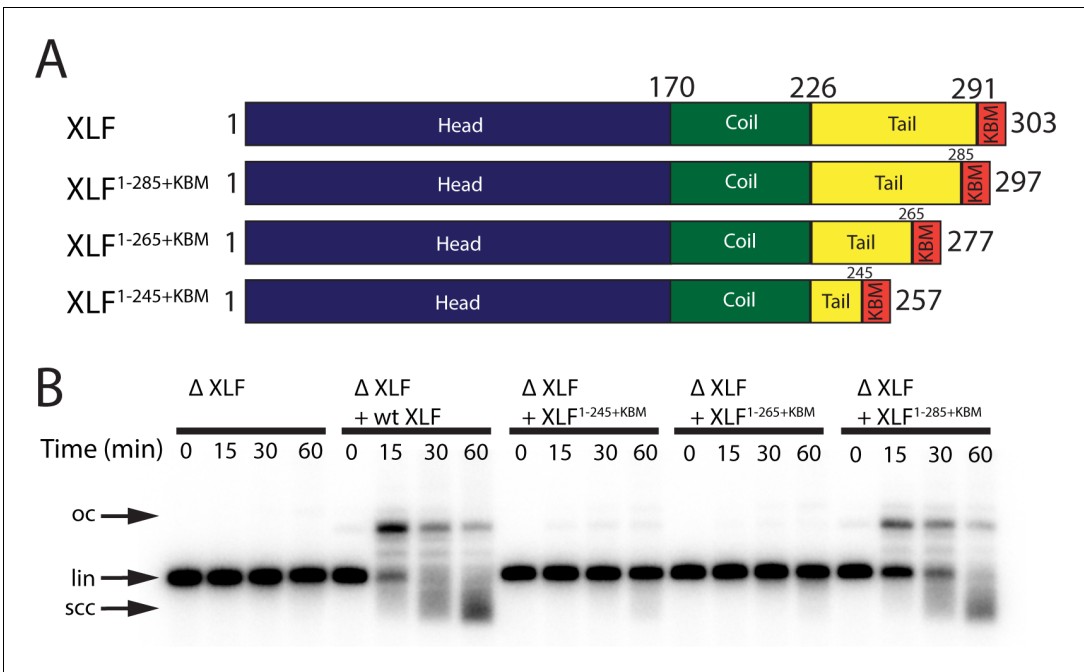

**Figure 2.** The tail region of XLF is essential for DNA end joining. (**A**) Schematics of the domain organization of wt XLF and truncation mutants (XLF[1-285+KBM], XLF[1-265+KBM], and XLF[1-245+KBM]). The residue number at the boundary of each region of the protein is indicated. (**B**) Ensemble time course end joining assay in XLF-depleted *Xenopus* egg extract. Recombinant wt XLF, XLF[1-285+KBM], XLF[1-265+KBM], and XLF[1-245+KBM] were added back at 75 nM final concentration to their respective reaction samples. DNA species: scc, supercoiled closed circular; lin, linear; oc, open circle.

The online version of this article includes the following source data and figure supplement(s) for figure 2:

**Figure supplement 1.** Characterization of XLF truncation mutants.

**Figure supplement 1—source data 1.** Source data for graph shown in *Figure 2—figure supplement 1B*.

the unstructured tail of XLF is necessary for NHEJ, we examined the ability of each C-terminal truncation mutant to rescue end joining in XLF-depleted extract (*Figure 2B*). At 75 nM the more severe truncations, XLF[1-265+KBM] and XLF[1-245+KBM], showed no significant joining activity, while the most conservative truncation, XLF[1-285+KBM], was able to rescue end joining. Even at a concentration of 500 nM, joining by XLF[1-245+KBM] was barely detectable while joining by XLF[1-265+KBM] was clearly much slower than the wild type. In contrast, joining by XLF[1-285+KBM] was similar to the wild type (*Figure 2— figure supplement 1A*). To ensure that these defects in end joining did not arise due to protein misfolding, we measured the stability of each mutant using differential scanning fluorimetry (*Figure 2— figure supplement 1B*). Consistent with all three mutants being stably folded, we found that the melting temperatures ($T_m$) of the mutants were similar to the $T_m$ of wt XLF. Thus, the tail of XLF is required for efficient end joining in egg extract.

## The interior region of the XLF C-terminal tail is required to form SR synaptic complex

To further interrogate how truncating the tail of XLF leads to defects in end joining, we used a smFRET assay that reports on the formation of the SR complex. This assay utilizes a 2 kb linear DNA labeled with Cy3 and Cy5 dyes seven nt from each blunt end of the substrate and contains an internal biotin that is used to immobilize it on the surface of a flow cell (*Figure 3A*; *Graham et al., 2016*; *Graham et al., 2017*). FRET between Cy3 and Cy5 only occurs within the SR complex with the average FRET efficiency being indistinguishable from the ligated product (*Graham et al., 2016*). For these experiments, extract depleted of XLF was supplemented with either buffer or recombinant protein and flowed into the flow cell. Each replicate consisted of a movie of three fields of view (FOVs) imaged for 15 min each. Example trajectories from the ΔXLF + wt XLF condition are shown in

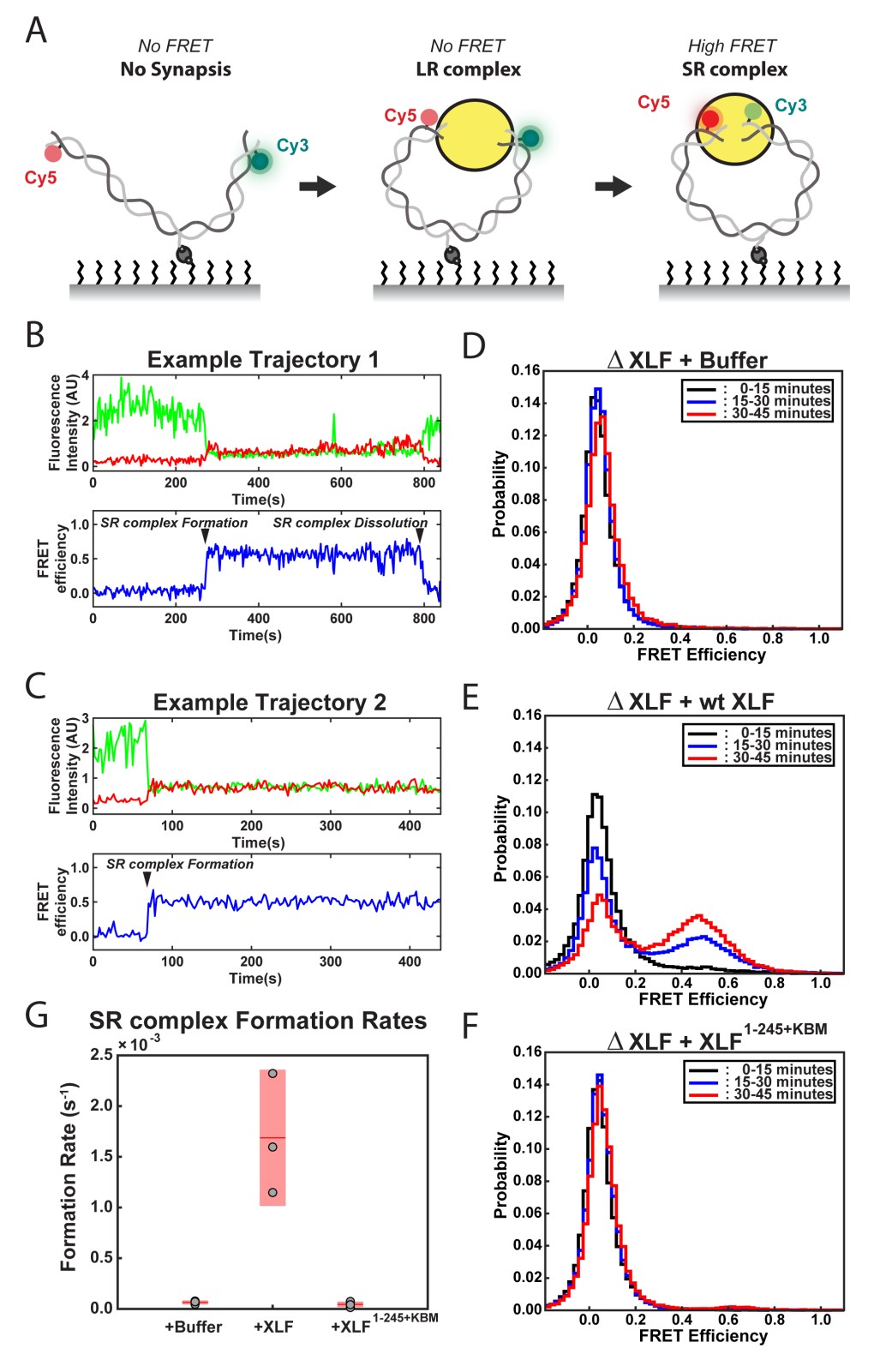

**Figure 3.** DNA end synapsis requires the tail of XLF. (**A**) Schematic of the FRET-labeled DNA substrate immobilized in the flow cell via biotin-streptavidin interactions. In the absence of egg extract the DNA substrate exhibits no FRET. When the LR complex forms, the DNA ends are not positioned close enough together for energy transfer from Cy3 to Cy5 even though the ends are co-localized within the NHEJ synaptic complex (*Graham et al., 2016*). Upon formation of the SR complex, FRET between the fluorophores on opposing DNA ends can be observed. (**B,C**) Example

*Figure 3 continued on next page*

*Figure 3 continued*

trajectories that contain SR complex formation events. Donor and acceptor fluorescence intensity are shown in green and red, respectively. The corresponding FRET efficiency from each trace is shown in blue in a separate trajectory below. (D, E, F) Normalized FRET histograms for each experimental condition accumulated over a 15-min time window. Data from each 15 min field of view is represented by a separate curve. (G) Plot of SR complex formation rates. For each condition, individual replicates are plotted as gray circles, and the mean is represented as dark red horizontal line. The 95% confidence interval is represented for each condition as a light red rectangle centered on the mean. This plot was generated using the notBoxPlot MATLAB function (*Campbell, 2020*).

The online version of this article includes the following source data and figure supplement(s) for figure 3:

**Source data 1.** Source data for histograms shown in *Figure 3D–F*.
**Source data 2.** Source data for plot shown in *Figure 3G*.
**Figure supplement 1.** Example smFRET trajectories.

*Figure 3B–C*. In these trajectories two distinct FRET states are observed: a low- or no-FRET state corresponding to unpaired DNA ends or the LR complex and a high-FRET state that corresponds to the SR complex and ultimately the ligated DNA product. *Figure 3B* shows an example trajectory where the SR complex forms and subsequently falls apart. *Figure 3C* shows an example where SR complex formation leads to the high-FRET state that persists until the end of the observation window. In *Figure 3D–F*, the FRET signal is plotted as a normalized histogram for each 15 min interval of the experiment. In the case of the ΔXLF + wt XLF condition, we observed a time-dependent increase in the high-FRET population (FRET efficiency ~ 0.5) due to accumulation of the SR complex along with ligated products (*Figure 3E*). In contrast, the high-FRET population was not observed in the ΔXLF + buffer (*Figure 3D* and *Figure 3—figure supplement 1A*) or ΔXLF + XLF$^{1-245+KBM}$ (*Figure 3F* and *Figure 3—figure supplement 1B*) conditions. The absence of a high-FRET population in the XLF$^{1-245+KBM}$ condition could be due to (1) a substantial decrease in the stability of the SR complex or (2) an inability of the SR complex to form. To distinguish between these possibilities, we measured the rate of SR complex formation by recording the number of individual FRET events detected from all single molecules tracked (*Supplementary file 1*). The SR complex formation rate for wt XLF ($1.7 \times 10^{-3} \pm 3.4 \times 10^{-4}$ s$^{-1}$) agrees well with a previously published value (*Graham et al., 2018*), while the rates for the buffer and XLF$^{1-245+KBM}$ conditions were > 25 fold lower, at $6.4 \times 10^{-5} \pm 1.0 \times 10^{-5}$ s$^{-1}$ and $4.6 \times 10^{-5} \pm 1.3 \times 10^{-5}$ s$^{-1}$, respectively (*Figure 3G*). As we observed a similarly low number of high FRET events (SR complex formation) for the buffer and XLF$^{1-245+KBM}$ conditions (*Supplementary file 1*), these results show that XLF$^{1-245+KBM}$ is deficient in forming the SR complex. Given the severe defect in SR complex formation rates for these conditions, we were unable to collect enough events to compare the stability of the SR complex. Collectively, these results indicate that the tail of XLF is required for formation of the SR complex and the close alignment of DNA ends.

## The sequence of the C-terminal tail of XLF is not critical for its role in end joining

The synapsis and joining defects seen for the XLF truncation mutants suggest two potential roles for the C-terminal tail of XLF: (1) the length of the XLF tail may be critical within the architecture of the NHEJ synaptic complex, allowing XLF to interact with binding partners Ku80 and XRCC4; alternatively, (2) residues within the C-terminal tail may be critical sites of phosphorylation or make contacts with other factors. Both DNA-PKcs and ATM are known to target sites within the XLF tail for phosphorylation (*Yu et al., 2008*). To test whether phosphorylation sites within the tail are required for end joining, we mutated all serines to glycines and all threonines to alanines between residues 226 and 292. This mutant, XLF$^{NoPhos+KBM}$, was able to rescue end joining in XLF-depleted extract as efficiently as wt XLF (*Figure 4A*), consistent with XLF phosphorylation not being required for end synapsis and ligation. To disrupt any unidentified interaction motifs, we randomly shuffled the sequence between residues 226 and 292 of XLF (*Figure 4—figure supplement 1A*). Sequences were shuffled using the Sequence Manipulation Suite, whereas the KBM was left unaltered because it is necessary for end joining (*Figure 1C*; *Stothard, 2000*). We constructed two distinct shuffled mutants using this approach, XLF$^{ShuffA+KBM}$ and XLF$^{ShuffB+KBM}$ (*Figure 4—figure supplement 1A*), and each of these mutants rescued end joining in XLF-depleted extract with wild type-like efficiency (*Figure 4A*).

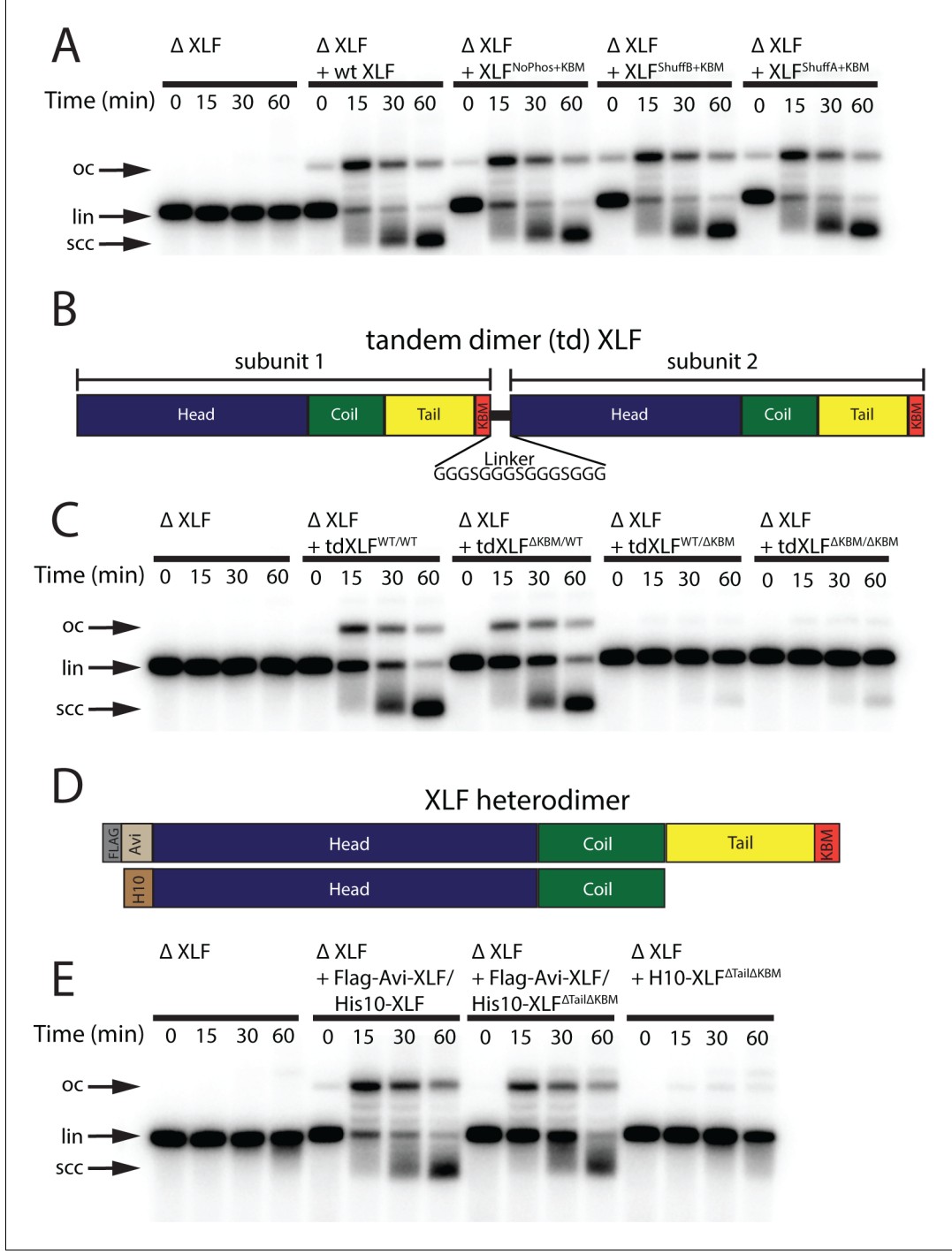

**Figure 4.** Requirements of the XLF tail in end joining. (**A**) Ensemble time course end joining assay in XLF-depleted *Xenopus* egg extract. Recombinant wt XLF, XLF$^{NoPhos+KBM}$, XLF$^{ShuffB+KBM}$, and XLF$^{ShuffA+KBM}$ were added back at 75 nM to their respective reaction samples. DNA species: scc, supercoiled closed circular; lin, linear; oc, open circle. (**B**) Schematic of the tandem dimer (td) XLF construct. (**C**) Ensemble time course end joining assay in XLF-depleted *Xenopus* egg extract. Recombinant tdXLF$^{WT/WT}$, tdXLF$^{\Delta KBM/WT}$, tdXLF$^{WT/\Delta KBM}$, and tdXLF$^{\Delta KBM/\Delta KBM}$ were added back at 50 nM final concentration to their respective reaction samples. DNA species: scc, supercoiled closed circular; lin, linear; oc, open circle. (**D**) Schematics of the individual subunits coexpressed to form the XLF heterodimer, Flag-Avi-XLF/His10-XLF$^{\Delta tail\Delta KBM}$. (**E**) Ensemble time course end joining assay in XLF-depleted *Xenopus* egg extract. Recombinant heterodimers (Flag-Avi-XLF/His10-XLF and Flag-Avi-XLF/His10-XLF$^{\Delta tail\Delta KBM}$)

*Figure 4 continued*

and His10-XLF$^{\Delta tail\Delta KBM}$ were added back at 75 nM final concentration to their respective reaction samples. DNA species: scc, supercoiled closed circular; lin, linear; oc, open circle.

The online version of this article includes the following source data and figure supplement(s) for figure 4:

**Figure supplement 1.** Sequence alignment of XLF shuffled tail mutants.
**Figure supplement 2.** Characterization of the tdXLF and XLF heterodimer constructs.
**Figure supplement 3.** SR complex formation by XLF heterodimer constructs.
**Figure supplement 3—source data 1.** Source data for plot shown in *Figure 4—figure supplement 3A*.

---

Collectively, these results rule out the loss of phosphorylation sites or disruption of a required motif within the tail as the cause of the end joining defect exhibited by the truncations mutants.

## Only a single KBM and C-terminal tail is required for XLF to promote end joining in egg extract

Our results with XLF$^{NoPhos+KBM}$ and the shuffled tail mutants suggest that it is the length and not the sequence of the tail that is important for DNA end synapsis. Having previously shown that a single XLF dimer mediates synapsis (*Graham et al., 2018*), we considered whether each C-terminal tail of the XLF dimer must bind the Ku molecules located on opposing DNA ends of the DSB. If this network of interactions is critical for the formation of the SR complex, decreasing the length of the XLF tail or removing one of the KBMs from the XLF dimer should block end joining. To test this Ku-XLF-Ku bridge model, we constructed a tandem dimer of XLF (tdXLF), which allows for the introduction of asymmetric KBM mutations (*Graham et al., 2018*). In this construct, the two subunits of XLF were expressed as a single polypeptide connected by a flexible linker (*Figure 4B*).

We generated mutants of the tdXLF construct in which either the KBM in subunit 1 is replaced by additional flexible linker sequence (tdXLF$^{\Delta KBM/WT}$), the KBM in subunit 2 is deleted (tdXLF$^{WT/\Delta KBM}$), or both the KBM in subunit 1 is replaced by linker sequence and the KBM in subunit 2 is deleted (tdXLF$^{\Delta KBM/\Delta KBM}$) (*Figure 4B*). Robust end joining was observed when tdXLF$^{WT/WT}$ or tdXLF$^{\Delta KBM/WT}$ were used to rescue XLF-depleted egg extract, but little to no joining was observed with either tdXLF$^{WT/\Delta KBM}$ or tdXLF$^{\Delta KBM/\Delta KBM}$ (*Figure 4C*). The loss of end joining observed when we delete or mutate the C-terminal KBM within subunit 2, even if the KBM in subunit 1 is intact (*Figure 4B–C* and *Figure 4—figure supplement 2A*) is likely because the C-terminal residues of the XLF KBM bind Ku80 in an internal hydrophobic pocket with the –COO group at the C-terminus forming an electrostatic contact with Lys238 of Ku80 (in the human) (*Nemoz et al., 2018*). The combination of losing this electrostatic contact and having to sterically accommodate the flexible linker within the hydrophobic pocket likely inhibits the KBM of subunit 1 in the tandem dimer from binding Ku80 effectively. A similar trend was observed when we used tdXLF constructs containing a previously characterized point mutation within the KBM (Leu 301 to Glu) (*Figure 4—figure supplement 2A*). The homologous mutation in human XLF was shown to reduce the affinity of XLF for Ku ~ 5-fold in vitro and impaired recruitment to DNA damage in cells (*Nemoz et al., 2018*). Robust joining was observed when tdXLF$^{L301E/WT}$ was used to rescue the XLF depletion, but little to no joining occurred when rescuing with tdXLF$^{WT/L301E}$ or tdXLF$^{L301E/L301E}$ (*Figure 4—figure supplement 2A*). Since we see robust joining with tdXLF$^{\Delta KBM/WT}$ and tdXLF$^{L301E/WT}$, these results demonstrate that a single KBM within an XLF dimer is sufficient for XLF's role in promoting end joining in extracts.

Although the above results rule out a Ku-XLF-Ku bridge mediated by both KBMs of an XLF dimer, they do not address the contributions by the tails outside of the KBM. To that end, we purified a XLF heterodimer in which one tail was deleted including the KBM. Heterodimers of XLF were generated by simultaneously expressing two versions of XLF, one His tagged and the other Flag-Avi tagged (*Figure 4D*; *Graham et al., 2018*). Subsequent tandem affinity purification allows for the isolation of the heterodimer, as it is the only species with both affinity tags. Similar to full-length XLF heterodimers (*Graham et al., 2018*), we verified that there is little to no exchange of subunits after expression and purification (*Figure 4—figure supplement 2B–C*). Size-exclusion chromatography and multi-angle light scattering (SEC-MALS) experiments confirmed that these heterodimer constructs exist as dimers in solution (*Figure 4—figure supplement 2D*). A comparison of joining efficiency for the Flag-Avi-XLF/His10-XLF$^{\Delta tail\Delta KBM}$ construct and the wild-type Flag-Avi-XLF/His10-XLF

revealed that both constructs can rescue end joining in XLF-depleted extracts, whereas dimers of His10-XLF$^{\Delta tail\Delta KBM}$ cannot (*Figure 4E*). However, the linear substrate persisted at later times in the Flag-Avi-XLF/His10-XLF$^{\Delta tail\Delta KBM}$ rescue relative to Flag-Avi-XLF/His10-XLF (*Figure 4E*). Consistent with these observations, a comparison of these same constructs using the smFRET SR complex formation assay showed that while both Flag-Avi-XLF/His10-XLF$^{\Delta tail\Delta KBM}$ and Flag-Avi-XLF/His10-XLF form the SR complex, Flag-Avi-XLF/His10-XLF$^{\Delta tail\Delta KBM}$ does so less efficiently (*Figure 4—figure supplement 3A–D*). We expect this defect is due to decreased XLF recruitment, but we cannot rule out other mechanisms by which end joining could be enhanced by the presence of two full-length C-terminal tails. Overall, these results demonstrate that a single KBM on a single tail of XLF is sufficient for robust end joining.

## The C-terminal tail of XLF must be sufficiently long to interact with XRCC4 within the synaptic complex

We next asked whether the tail is required for XLF to interact with other core NHEJ factors within the synaptic complex. Previously, we showed that the XLF-XRCC4 interaction is required for SR complex formation (*Graham et al., 2018*). The flexible tail may have to be of sufficient length to facilitate the interaction between the N-terminal head domains of XLF and XRCC4. To test this hypothesis, we performed DNA pulldown experiments in egg extract to evaluate the recruitment and stability of core NHEJ factors to DNA ends. DNA containing a biotin at both 5′ ends was attached to magnetic beads and either cut to introduce a blunt-ended DSB or left intact to control for non-specific DNA binding (*Figure 5A*). After being incubated in *Xenopus* egg extract, the beads were isolated and washed. The stably associated proteins were identified by western blot. Robust recruitment of the core NHEJ factors Ku, DNA-PKcs, XRCC4, Lig4, and XLF was observed in the mock-depleted extract relative to the uncut control DNA (*Figure 5B*). Consistent with previous reports, the lower electrophoretic mobility of XRCC4 and XLF is likely due to phosphorylation by DNA-PKcs (*Cottarel et al., 2013*; *Yu et al., 2008*).

To determine if XLF is required to stabilize XRCC4-Lig4 at DNA ends, we depleted XLF from egg extract and then blotted for NHEJ factors in our pulldown assay. Loss of XLF led to a large reduction in XRCC4 and Lig4 signal that could be rescued by addition of recombinant wild type XLF. As XRCC4-Lig4 is known to be recruited in the absence of XLF (*Wu et al., 2007*), these results are consistent with the interaction between XLF and XRCC4 stabilizing the XRCC4-Lig4 complex at DNA ends. We next tested XLF$^{L117D}$ (human XLF$^{L115D}$), an XRCC4 interaction deficient mutant (*Graham et al., 2018*). XLF$^{L117D}$ was associated with DNA ends yet could not restore XRCC4-Lig4 stability (*Figure 5B*). Similarly, XLF$^{1-245+KBM}$ failed to restore XRCC4-Lig4 stability even though it was robustly recruited. Collectively, these results show that a minimal tail length is necessary for XLF to stabilize the XRCC4-Lig4 complex.

## NHEJ in cells also requires a single KBM and a sufficiently long C-terminal tail

Our observations in egg extract define the requirements of the XLF C-terminal tail during NHEJ. To determine if the same interactions are important in cells, we used a chromosomal end joining assay (EJ7-GFP) that reports on error-free repair of DSBs induced by Cas9 and single-guide RNAs (sgRNAs) (*Bhargava et al., 2018*). In this assay, a green fluorescent protein (GFP) expression cassette, with a 46 nt insert that disrupts GFP, is integrated into the *Pim1* locus in *Xlf* $^{-/-}$ mouse embryonic stem cells (mESCs). Two tandem DSBs induced by Cas9/sgRNAs excise this insert, and subsequent error-free repair between the distal DSB ends restores GFP. Thus, determining the % GFP+ cells provides the frequency of this end joining event, which is normalized to transfection efficiency (see Materials and methods). With this assay, error-free end joining depends solely on canonical NHEJ factors (*Figure 6A*; *Bhargava et al., 2018*). We transfected various human XLF constructs into *Xlf*$^{-/-}$ mESCs, along with the Cas9/sgRNA plasmids (*Figure 6—figure supplement 1A*), and measured the percentage of GFP+ cells. As previously observed, GFP+ cells are dependent on XLF in that transfections with wt XLF causes a high frequency of GFP+ cells (~50%), but without XLF (i.e. empty vector) GFP+ cells are near background levels (*Figure 6B*). Similar to our results in *Xenopus* egg extract we observed that the XLF KBM and tail were both important for efficient end joining in cells (*Figure 6B*). As compared to wt XLF, XLF$^{\Delta KBM}$ led to a severe joining defect (sixfold), and XLF$^{1-}$

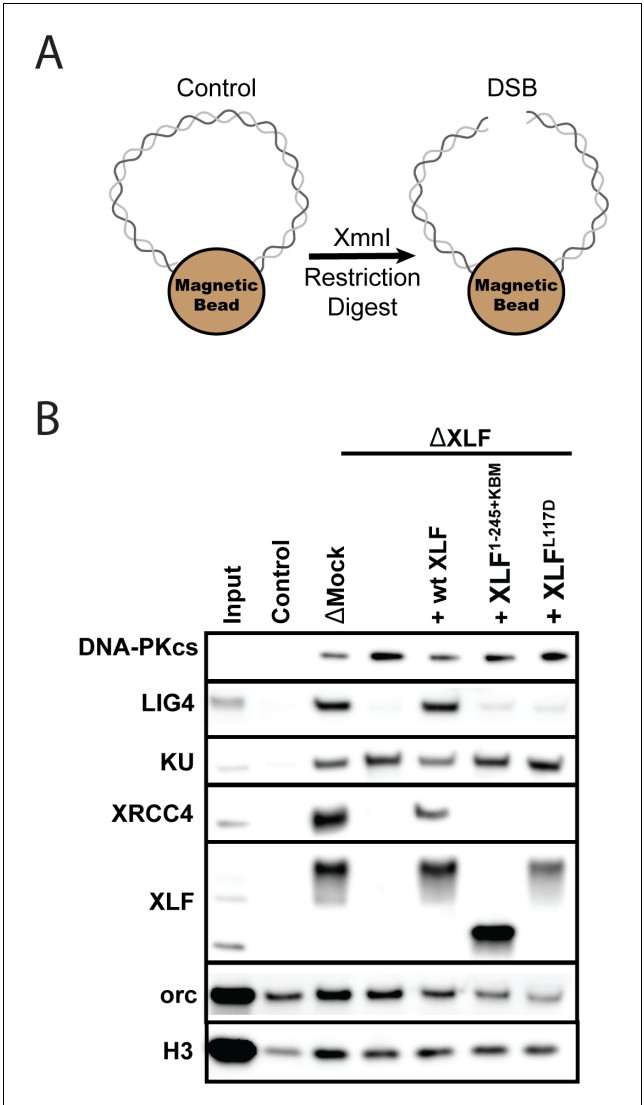

**Figure 5.** The XLF tail is required to stabilize XRCC4-Lig4. (**A**) Cartoon schematic of the DNA pulldown assay. Both ends of a linear DNA substrate are conjugated to magnetic beads, and either cut with XmnI to generate a DSB with blunt ends or left uncut as a control. (**B**) Immunoblots of NHEJ core factors (DNA-PKcs, Lig4, Ku, XRCC4, and XLF) and the loading controls, Orc and H3, bound to DNA-beads after a 15-min incubation in egg extract. Samples run in parallel were the input (extract diluted 1:40) and control (uncut DNA substrate pulldown) as well as pulldowns with the DSB substrate in either mock-depleted extract or XLF-depleted extract with recombinant wt XLF, XLF$^{1-245+KBM}$, or XLF$^{L117D}$ added back at 20 nM. The lower band observed in the XLF input sample is an unidentified cross-reactive species (see *Figure 1—figure supplement 1B*) that is not pulled down under the control or the DSB condition.

243+KBM led to a substantial reduction (3.2-fold). Next, we introduced tdXLF constructs into *Xlf⁻/⁻* mESCs to test whether a single KBM is sufficient for XLF function. Consistent with our results in egg extract, transfection of tdXLF$^{(WT/WT)}$ and tdXLF$^{(\Delta KBM/WT)}$ rescued repair as efficiently as wt XLF. However, tdXLF$^{(\Delta KBM/\Delta KBM)}$ showed a severe defect compared to wt XLF (12.2-fold). Overall, these results demonstrate that a single KBM within an XLF dimer is sufficient for end joining in cells, but the tail of XLF must be sufficiently long to facilitate efficient end joining.

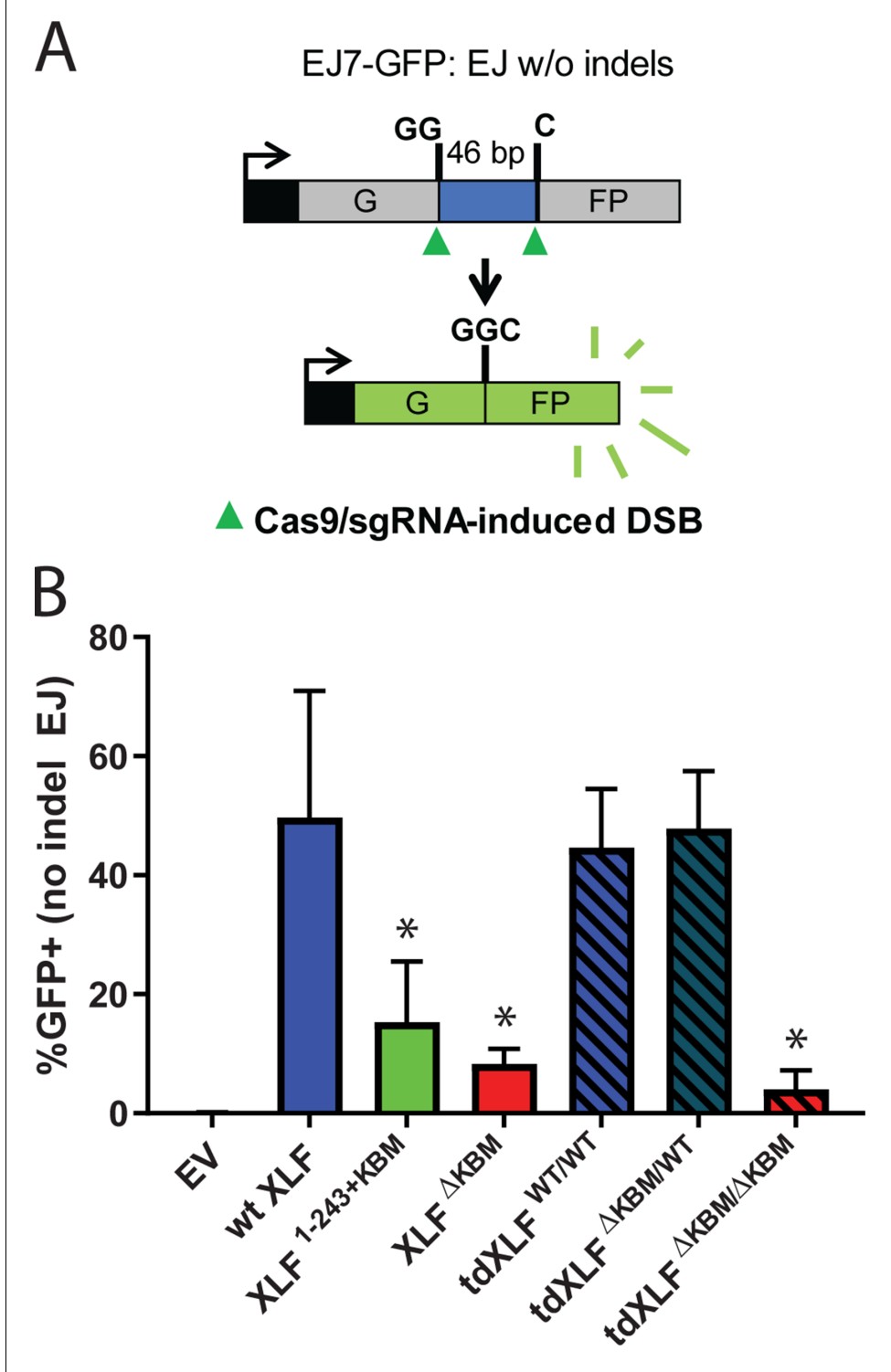

**Figure 6.** A single XLF tail is required for end joining in cells. (**A**) Schematic of the cellular GFP NHEJ reporter (EJ7-GFP). A 46 bp insertion is located within a region of the GFP gene that is critical for fluorescence. Cas9 and guide RNAs are expressed so that DSBs are induced on either end of this insertion. Fluorescence is restored only if the blunt ends of the GFP gene are repaired via error-free end joining. (**B**) Xlf[-/-] mESCs were transfected with an empty vector or the same vector containing a human XLF construct. The GFP frequencies were normalized against parallel transfections of a GFP+ expression vector. The normalized mean %GFP+ and corresponding standard deviation for each condition are shown. N=6 for each condition, and an unpaired T-Test with the Holm-Sidak

*Figure 6 continued on next page*

*Figure 6 continued*

correction was used to determine significance. An * represents datasets that are significantly different (p≤0.015) from the wt XLF results (XLF$^{1-243+KBM}$p=0.015, XLF$^{\Delta KBM}$p=0.003, tdXLF$^{WT/WT}$p=0.085, tdXLF$^{\Delta KBM/WT}$p=0.085, tdXLF$^{\Delta KBM/\Delta KBM}$p=0.002).

The online version of this article includes the following source data and figure supplement(s) for figure 6:

**Source data 1.** Source data for graph shown in *Figure 6B*.
**Figure supplement 1.** XLF constructs.

## Discussion

Synapsis of DNA ends during NHEJ is an essential but poorly understood process. Among the core NHEJ factors, the role of XLF in end joining has remained particularly elusive. Using *Xenopus* egg extract to recapitulate physiological end joining, our findings further articulate the intermolecular interactions formed between XLF and other NHEJ factors that mediate end synapsis. We show that both the XLF KBM and the tail are necessary for end joining. Notably, the length of the XLF tail, and not its sequence, is important for NHEJ. We propose that the C-terminal tail of a single XLF monomer acts as a 'leash' that tethers XLF to Ku via its KBM and allows it to form required interactions with XRCC4 within the synaptic complex.

### The role of the XLF Ku-binding motif in NHEJ

Recruitment of NHEJ factors to DSBs depends largely on Ku (*Frit et al., 2019*), and XLF is strictly required for end joining in *Xenopus* egg extract (*Graham et al., 2016*). Here, we show that deleting the XLF KBM also largely ablates end joining (*Figure 1C*). This defect is likely due to a loss of XLF localization to DSBs, as has been observed in cells upon KBM deletion (*Yano et al., 2011*). Therefore, the KBM recruits XLF to DNA ends, which facilitates formation of the synaptic complex.

Similar to our results in egg extract, we found that introduction of XLF$^{\Delta KBM}$ into cells leads to a large decrease in error-free NHEJ, although end joining remains higher than in cells lacking XLF (*Figure 6B*). As end joining could be partially rescued by high concentrations of XLF$^{\Delta KBM}$ in egg extract (*Figure 1D*), these results suggest that XLF does not need the KBM to promote end joining when the local XLF concentration is high enough. Altogether, these findings indicate that the XLF KBM is critical for NHEJ, although without the KBM, XLF retains a weak residual activity that is likely mediated through its interaction with XRCC4.

### Recruitment of XLF to DSBs is necessary but not sufficient for NHEJ

Recruitment of XLF to DSBs is required for end joining but is not sufficient. In addition to the KBM, we show that the tail is critical for DNA end synapsis (*Figure 3G*) and subsequent end joining in egg extract (*Figure 2B*). Similarly, introducing an XLF mutant with a truncated tail that maintained the KBM (human XLF$^{1-243+KBM}$) resulted in a substantial drop in NHEJ efficiency in cells (*Figure 6B*). Prior to this study, the role of the KBM and the rest of the C-terminal tail had not been analyzed systematically. Structural studies of XLF have commonly deleted both the tail and KBM (*Li et al., 2008*; *Andres et al., 2007*; *Andres et al., 2012*; *Ropars et al., 2011*; *Hammel et al., 2011*) to facilitate crystallization (*Andres et al., 2007*; *Andres et al., 2012*). This construct still interacts with XRCC4 and promotes minimal end joining in in vitro reconstitution assays (*Andres et al., 2007*; *Hammel et al., 2011*). Within human cells, the XLF C-terminal tail was reported to be dispensable for V(D)J recombination (*Malivert et al., 2009*). As we observe a substantial reduction in NHEJ efficiency in mESCs upon removal of either the tail or KBM, this result may reflect a differential requirement of intermolecular interactions during V(D)J recombination as compared to spontaneous DSB repair. Our results demonstrate that both the flexible C-terminal tail and the KBM are required for canonical NHEJ under physiological conditions.

How does the tail of XLF contribute to NHEJ? Experiments in which we shuffled the sequence of the tail demonstrated that outside the KBM there are no additional motifs that are required for end joining (*Figure 4A*). Similarly, ablating all known and potential phosphorylation sites within this region did not affect end joining (*Figure 4A*), consistent with prior results (*Yu et al., 2008*; *Normanno et al., 2017*). These observations suggest that the length of the tail is the critical requirement for NHEJ and informs models of how XLF promotes end joining. Therefore, we considered a

model in which the two tails of the XLF dimer enable the formation of a Ku-XLF-Ku bridge (*Frit et al., 2019*) that is required to synapse DNA ends. Using synthetic tandem dimers of XLF, we generated asymmetric XLF dimers to show that a single KBM within an XLF dimer is sufficient to promote NHEJ in both *Xenopus* egg extract (*Figure 4B–C*) and in mammalian cells (*Figure 6B*). These results point toward a model where a single XLF-Ku contact is sufficient for end joining.

Alternating XLF and XRCC4 filaments have been proposed to synapse DNA ends and the tail of XLF has been implicated in stabilizing these filaments in vitro (*Andres et al., 2012*; *Hammel et al., 2011*). We disfavor this model as we have previously shown that these filaments do not form during NHEJ in egg extract (*Graham et al., 2018*). While minimal mixtures of *Xenopus* XLF and XRCC4 form large aggregates that bridge DNA – similar to their human homologs – quantitative single-molecule imaging showed that a single XLF dimer colocalizes with DNA shortly before the formation of the SR complex. Furthermore, significant numbers of XLF dimers fail to accumulate on DNA indicating that filament formation does not occur during any stage of the repair reaction. Recent work in a

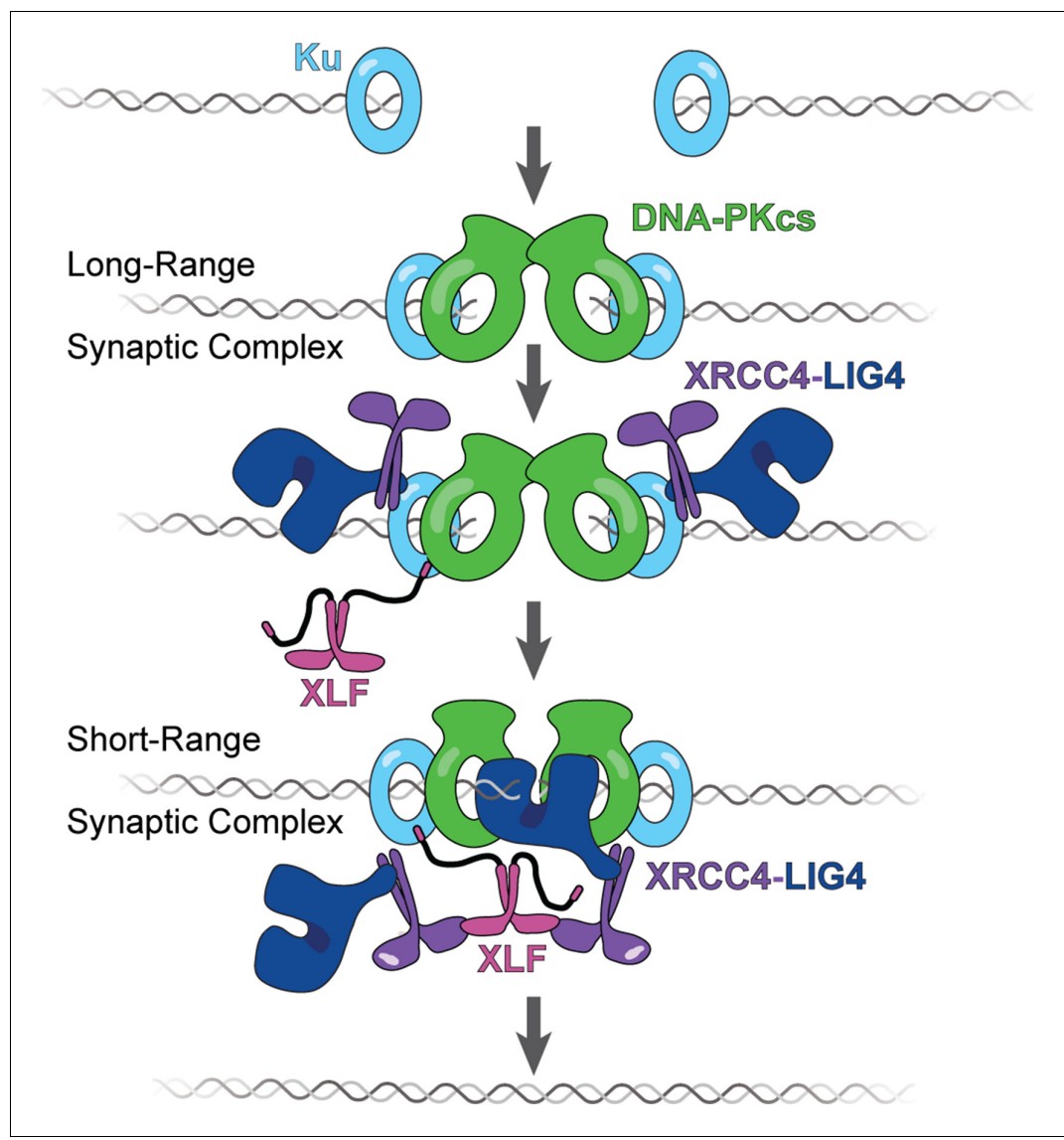

**Figure 7.** The tail enables XLF to stabilize XRCC4-Lig4 in the short-range (SR) complex. Cartoon model representing the evolution of the NHEJ synaptic complex. Ku initially binds DNA ends and DNA-PKcs is recruited shortly after to mediate formation of the long-range (LR) complex. The XLF KBM tethers it to Ku while the other domains of XLF can diffuse locally to find and bind XRCC4. This mediates the formation of the XRCC4-XLF-XRCC4 bridge that leads to SR complex formation and likely puts Lig4 in position to engage the DNA ends.

human reconstitution of NHEJ also suggests that XLF filaments are not necessary for end synapsis (*Zhao et al., 2019*). During repair, Lig4, which is required to recruit and stabilize XRCC4 at DNA ends (*Costantini et al., 2007*; *Wu et al., 2007*), likely blocks XLF molecules from interacting with both XRCC4 monomers thereby preventing filament formation (*Recuero-Checa et al., 2009*; *Ochi et al., 2012*).

## A model for XLF in DNA end synapsis during NHEJ

Our data supports a sequential model that describes how XLF facilitates formation of the NHEJ SR synaptic complex (*Figure 7*). A single XLF dimer is first recruited to Ku via its KBM. Subsequently, the tail enables XLF to find and engage with other binding partners within the complex. In this way, the tail acts as a flexible connector that allows Ku-bound XLF to diffuse locally and make additional contacts that promote the transition from the LR to SR complex. One of these potential interactions is between XLF and XRCC4, an interaction that is required for NHEJ in cells (*Malivert et al., 2010*; *Roy et al., 2015*; *Roy et al., 2012*). We have previously shown that a single XLF dimer must engage two XRCC4-Lig4 complexes, one through each head domain, for efficient SR complex formation and end joining (*Graham et al., 2018*). Here, we demonstrate that a sufficient XLF tail length is required for both SR complex formation (*Figure 3G*) and for XLF to stabilize XRCC4-Lig4 at DNA ends (*Figure 5B*). These results mimic XLF$^{L117D}$, an XLF mutant deficient in interaction with XRCC4 (*Graham et al., 2018*). We propose that the tail of XLF facilitates the formation of an XLF-XRCC4 bridge that spans the DNA break or positions XRCC4-Lig4 so that Lig4 can engage the DNA ends. These models need not be mutually exclusive and future studies will be needed to elucidate the structural role of both XRCC4 and Lig4 in end synapsis during NHEJ.

## Materials and methods

### Key resources table

| Reagent type (species) or resource | Designation | Source or reference | Identifiers | Additional information |
|---|---|---|---|---|
| Antibody | Anti-XLF (rabbit polyclonal) | New England Peptide, *Graham et al., 2016* DOI: 10.1016/j.molcel.2016.02.010 | Loparo Lab NEP 2993 | See Materials and methods, Immuno depletion. (1:500) |
| Recombinant DNA reagent | His10-SUMO-xl XLF | *Graham et al., 2016* DOI: 10.1016/j.molcel.2016.02.010 | Loparo Lab pTG296 | See Materials and methods, Plasmid Construction, XLF and XLF Truncation Mutants |
| Recombinant DNA reagent | His10-SUMO-xl XLF$^{\Delta KBM}$ | This work | Loparo Lab pSC49 | See Materials and methods, Plasmid Construction, XLF and XLF Truncation Mutants |
| Recombinant DNA reagent | His10-SUMO-xl XLF$^{1-245+KBM}$ | This work | Loparo Lab pSC26 | See Materials and methods, Plasmid Construction, XLF and XLF Truncation Mutants |
| Recombinant DNA reagent | His10-SUMO-xl XLF$^{1-265+KBM}$ | This work | Loparo Lab pSC32 | See Materials and methods, Plasmid Construction, XLF and XLF Truncation Mutants |
| Recombinant DNA reagent | His10-SUMO-xl XLF$^{1-285+KBM}$ | This work | Loparo Lab pSC25 | See Materials and methods, Plasmid Construction, XLF and XLF Truncation Mutants |
| Recombinant DNA reagent | His10-SUMO-xl XLF$^{NoPhos+KBM}$ | This work | Loparo Lab pSC48 | See Materials and methods, Plasmid Construction, XLF NoPhos and Shuffled Tail Mutants |

*Continued on next page*

*Continued*

| Reagent type (species) or resource | Designation | Source or reference | Identifiers | Additional information |
|---|---|---|---|---|
| Recombinant DNA reagent | His10-SUMO-xl XLF$^{ShuffB+KBM}$ | This work | Loparo Lab pSC41 | See Materials and methods, Plasmid Construction, XLF NoPhos and Shuffled Tail Mutants |
| Recombinant DNA reagent | His10-SUMO-xl XLF$^{ShuffA+KBM}$ | This work | Loparo Lab pSC40 | See Materials and methods, Plasmid Construction, XLF NoPhos and Shuffled Tail Mutants |
| Recombinant DNA reagent | His10-SUMO-xl XLF$^{WT/WT}$ | *Graham et al., 2018* DOI:10.1038/s41594-018-0120-y | Loparo Lab pTG454 | See Materials and methods, Plasmid Construction, tdXLF constructs |
| Recombinant DNA reagent | His10-SUMO-xl tdXLF$^{ΔKBM/WT}$ | This work | Loparo Lab pSC64 | See Materials and methods, Plasmid Construction, tdXLF constructs |
| Recombinant DNA reagent | His10-SUMO-xl tdXLF$^{WT/ΔKBM}$ | This work | Loparo Lab pSC63 | See Materials and methods, Plasmid Construction, tdXLF constructs |
| Recombinant DNA reagent | His10-SUMO-xl tdXLF$^{ΔKBM/ΔKBM}$ | This work | Loparo Lab pSC76 | See Materials and methods, Plasmid Construction, tdXLF constructs |
| Recombinant DNA reagent | His10-SUMO-xl tdXLF$^{L301E/WT}$ | This work | Loparo Lab pSC65 | See Materials and methods, Plasmid Construction, tdXLF constructs |
| Recombinant DNA reagent | His10-SUMO-xl tdXLF$^{WT/L301E}$ | This work | Loparo Lab pSC66 | See Materials and methods, Plasmid Construction, tdXLF constructs |
| Recombinant DNA reagent | His10-SUMO-xl tdXLF$^{L301E/L301E}$ | This work | Loparo Lab pSC68 | See Materials and methods, Plasmid Construction, tdXLF constructs |
| Recombinant DNA reagent | His10-xl XLF: Flag-Avi-xl XLF | *Graham et al., 2018* DOI:10.1038/s41594-018-0120-y | Loparo Lab pTG448 | See Materials and methods, Plasmid Construction, XLF heterodimers |
| Recombinant DNA reagent | His10-xl XLF$^{ΔTailΔKBM}$: Flag-Avi-xl XLF | This work | Loparo Lab pSC52 | See Materials and methods, Plasmid Construction, XLF heterodimers |
| Recombinant DNA reagent | His10-xl XLF$^{ΔTailΔKBM}$ | This work | Loparo Lab pSC79 | See Materials and methods, Plasmid Construction, XLF heterodimers |
| Recombinant DNA reagent | His10-SUMO-xl XLF$^{L117D}$ | *Graham et al., 2018* DOI:10.1038/s41594-018-0120-y | Loparo Lab pTG339 | See Materials and methods, Plasmid Construction, XLF and XLF Truncation Mutants |
| Recombinant DNA reagent | pCAGGS-BSKX (Empty Vector) | *Bhargava et al., 2018* DOI: 10.1038/s41467-018-04867-5 | Stark Lab JS74 | |
| Recombinant DNA reagent | pCAGGS-BSKX 3xFlag-hXLF | This work | Loparo Lab pSC70 | See Materials and methods, Plasmid Construction, Human XLF and tdXLF constructs |
| Recombinant DNA reagent | pCAGGS-BSKX 3xFlag-hXLF$^{1-243-KBM}$ | This work | Loparo Lab pSC77 | See Materials and methods, Plasmid Construction, Human XLF and tdXLF constructs |
| Recombinant DNA reagent | pCAGGS-BSKX 3xFlag-hXLF$^{ΔKBM}$ | This work | Loparo Lab pSC78 | See Materials and methods, Plasmid Construction, Human XLF and tdXLF constructs |

*Continued on next page*

*Continued*

| Reagent type (species) or resource | Designation | Source or reference | Identifiers | Additional information |
|---|---|---|---|---|
| Recombinant DNA reagent | pCAGGS-BSKX 3x Flag-tdhXLF$^{WT/WT}$ | This work | Loparo Lab pSC71 | See Materials and methods, Plasmid Construction, Human XLF and tdXLF constructs |
| Recombinant DNA reagent | pCAGGS-BSKX 3x Flag-tdhXLF$^{\Delta KBM/WT}$ | This work | Loparo Lab pSC72 | See Materials and methods, Plasmid Construction, Human XLF and tdXLF constructs |
| Recombinant DNA reagent | pCAGGS-BSKX 3xFlag-tdhXLF$^{\Delta KBM/\Delta KBM}$ | This work | Loparo Lab pSC73 | See Materials and methods, Plasmid Construction, Human XLF and tdXLF constructs |
| Software, algorithm | *Source code 1* | *Graham et al., 2016*; DOI: 10.1016/j.molcel.2016.02.010 *Graham et al., 2018*; DOI: 10.1038/s41594-018-0120-y *Graham et al., 2017*; DOI: 10.1016/bs.mie.2017.03.020 | Loparo Lab circFRET_roving 4_drift_thumbs | See Materials and methods, Single-Molecule FRET SR Complex Formation Assay |
| Software, algorithm | *Source code 2* | *Graham et al., 2016*; DOI: 10.1016/j.molcel.2016.02.010 *Graham et al., 2018*; DOI: 10.1038/s41594-018-0120-y *Graham et al., 2017*; DOI: 10.1016/bs.mie.2017.03.020 | Loparo Lab simple_browselo calback_thumbnails | See Materials and methods, Single-Molecule FRET SR Complex Formation Assay |

## Plasmid construction

All expression plasmid constructs used are listed in the Key Resources Table. *xl* in the plasmid name denotes constructs based on the XLF sequence from *Xenopus laevis*. *h* in the plasmid name denotes constructs based on the human XLF sequence.

### XLF and XLF truncation mutants

The construction of the His10-SUMO-xl XLF (*Xenopus laevis* XLF) expression plasmid, pTG296, has been previously described (*Graham et al., 2016*). This plasmid was used as the template to generate the His10-SUMO-xl XLF$^{\Delta KBM}$, His10-SUMO-xl XLF$^{1-245+KBM}$, His10-SUMO-xl XLF$^{1-265+KBM}$, and His10-SUMO-xl XLF$^{1-285+KBM}$ *Xenopus laevis* XLF constructs in the same expression vector using round the horn mutagenesis (*Hemsley et al., 1989*). This method generates a linear PCR product where edits to the template are introduced at the termini. This linear product is then phosphorylated and ligated to produce a circular plasmid. The same approach was used to create the His10-SUMO-xl XLF$^{L117D}$ expression plasmid (*Graham et al., 2018*).

### XLF NoPhos and shuffled tail mutants

To generate His10-SUMO-xl XLF$^{NoPhos+KBM}$, a geneblock that contained the sequence corresponding to the *Xenopus laevis* XLF C-terminal tail region (amino acids between and including 226–292) was ordered (Integrated DNA Technologies) where all serine residues in this region were mutated to glycine and all threonine residues in this region were mutated to alanine. This geneblock was then inserted into the appropriate position to replace the original sequence corresponding to the C-terminal tail within the H10-SUMO-xl XLF expression plasmid by isothermal (Gibson) assembly (*Gibson et al., 2009*). The His10-SUMO-xl XLF$^{ShuffA+KBM}$ and His10-SUMO-xl XLF$^{ShuffA+KBM}$ constructs were generated using the same approach. The shuffled sequences of the C-terminal tail regions in these mutants were generated using the Protein Sequence Shuffle Tool within the Sequence Manipulation Suite (*Stothard, 2000*).

### tdXLF constructs

The construction of the tandem dimer of *Xenopus laevis* XLF, H10-SUMO-xl tdXLF$^{WT/WT}$, has been previously described (*Graham et al., 2018*). This tandem dimer consists of two XLF sequences connected by a linker composed of a repeating 'GGGS' amino acid sequence (*Graham et al., 2018*). To create the H10-SUMO-xl tdXLF$^{\Delta KBM/WT}$ construct, the KBM from the XLF subunit one sequence was

deleted and additional linker sequence (GGGSGGGSGGGSGGGS) was added to prevent defects due to shortening the flexible tail and linker region between subunit 1 and subunit 2. This was accomplished by amplifying the H10-SUMO-xl tdXLF$^{WT/WT}$ as two separate fragments and assembling them using isothermal (Gibson) assembly (*Gibson et al., 2009*). The H10-SUMO-xl tdXLF $^{WT/\Delta KBM}$ was created using the same method, but in this case the KBM of subunit two was deleted outright and not replaced. The double mutant, H10-SUMO-tdXLF $^{\Delta KBM /\Delta KBM}$, was also generated using this method but used H10-SUMO-xl tdXLF$^{\Delta KBM/WT}$ as the template. The same approach was used to create expression plasmids containing H10-SUMO-xl tdXLF $^{L301E/WT}$, H10-SUMO-xl tdXLF $^{WT/L301E}$, and H10-SUMO-xl tdXLF $^{L301E/L301E}$. No additional linker sequence was introduced for any of the L301E point mutants.

## XLF heterodimers
The procedure by which both His10-xl XLF and Flag-Avi-xl XLF (both *Xenopus laevis* XLF sequences) were cloned into a dual expression vector has been described previously (*Graham et al., 2018*). Round the horn mutagenesis was used to generate the Flag-Avi-xl XLF/His10-xl XLF$^{\Delta tail\Delta KBM}$ construct (*Hemsley et al., 1989*). The His10-xl XLF$^{\Delta tail\Delta KBM}$ construct was generated by the same method using the Flag-Avi-XLF/His10-XLF$^{\Delta tail\Delta KBM}$ expression plasmid as a template.

## Human XLF and tdXLF constructs
The cloning of the human XLF constructs into a pCAGGS-BSKX vector was performed as previously described (*Bhargava et al., 2018*). Briefly, each XLF construct was ordered as a geneblock from Integrated DNA Technologies with a 3xFlag tag at the N-terminus. The pCAGGS-BSKX vector was linearized by cutting with EcoR1-HF (New England Biolabs) and Xho1 (New England Biolabs). Each geneblock was then inserted into the linearized vector by isothermal (Gibson) assembly (*Gibson et al., 2009*). Similar to the construction of the *Xenopus* tandem dimer construct, a custom python script was used to generate distinct DNA sequences of human XLF for each XLF sequence included in the tandem dimer to facilitate cloning (*Graham et al., 2018*). These two distinct XLF sequences are separated by a 'GGGSGGGSGGGSGGG' linker.

## Protein purification
All purified recombinant proteins used are shown in *Figure 6—figure supplement 1B–C*.

### Xl XLF, XLF truncation mutants, XLF NoPhos, and shuffled tail mutants
His10-SUMO-xl XLF (wild type), His10-SUMO-xl XLF$^{1-245+KBM}$, His10-SUMO-xl XLF$^{1-265+KBM}$, His10-SUMO-xl XLF$^{1-285+KBM}$, His10-SUMO-xl XLF$^{NoPhos+KBM}$, His10-SUMO-xl XLF$^{ShuffA+KBM}$, His10-SUMO-xl XLF$^{ShuffB+KBM}$, and His10-SUMO-xl XLF$^{L117D}$ constructs were all purified using a previously detailed protocol (*Graham et al., 2018*). Each expression plasmid was transformed into *E. coli* BL21(DE3) pLysS cells. Cultures were grown at 37°C to an OD$_{600}$of ~ 0.6. IPTG was then added to cultures at a final concentration of 1 mM. The cultures were then moved to 25–30°C for 3 hr for protein expression. Cultures were then centrifuged to collect cells. Cell pellets were washed in 1x PBS buffer, flash frozen in liquid nitrogen, and stored at −80°C. The pellets were thawed and resuspended in 15 mL lysis buffer (20 mM Tris-HCl, pH 8.0, 1 M NaCl, 30 mM imidazole, 5 mM BME, and 1 mM PMSF) per liter of culture and sonicated. The resulting lysates were clarified by centrifugation for 1 hr at 20,000 rpm in a SS34 fixed angle rotor at 4°C. The supernatant was incubated with Ni-NTA agarose (Qiagen, Germantown, MD, USA) equilibrated in lysis buffer for 1 hr at 4°C to allow for binding of His-tagged proteins. The Ni-NTA resin was then was washed with lysis buffer followed by washing with salt reduction buffer (20 mM Tris-HCl, pH 8.0, 350 mM NaCl, 30 mM imidazole, 5 mM BME). Proteins bound to the Ni-NTA resin were then eluted by incubating the Ni-NTA resin in elution buffer (20 mM Tris-HCl, pH 8.0, 350 mM NaCl, 250 mM imidazole, 5 mM BME) for 2 min. This elution step was repeated several times. Peak fractions were pooled and dialyzed against His-SUMO dialysis buffer (20 mM Tris-HCl, pH 8.0, 350 mM NaCl, 10 mM imidazole, 5 mM BME, and 10% glycerol) at 4°C in the presence of H6-Ulp1 protease, which was added to cleave the H10-SUMO tags from the XLF constructs. After two rounds of dialysis, each being more than 4 hr long, the dialysate was incubated with fresh Ni-NTA resin equilibrated in dialysis buffer at 4°C for 1.5 hr. Any remaining H10-SUMO-protein, cleaved H10-SUMO, or H6-Ulp1 should remain bound to the Ni-NTA at this step.

The flow through which contains XLF protein lacking the H10-SUMO tag was collected and diluted in 1.33 volumes of SP Buffer (50 mM Na-MES, pH 6.5, 10% glycerol, 1 mM DTT) so that the [NaCl] becomes 150 mM, and subsequently passed over SP Sepharose Fast Flow (GE Healthcare (Milipore-Sigma), Burlington, MA, USA) equilibrated in SP Buffer A (50 mM Na-MES, pH 6.5, 150 mM NaCl, 10% glycerol, 1 mM DTT). The SP Sepharose Fast Flow resin was then washed with 10 column volumes of SP Buffer A. The protein was eluted with SP Buffer B (50 mM Na-MES, pH 6.5, 350 mM NaCl, 10% glycerol, 1 mM DTT) in one-column volume increments. All steps involving SP Sepharose Fast Flow were carried out at 4°C. Peak fractions were pooled, flash frozen in liquid nitrogen, and stored at −80°C until use.

The purification of His10-SUMO-xl XLF$^{\Delta KBM}$ followed the His10-SUMO purification protocol detailed above through the SUMO cleavage and dialysis step. The dialysate was spun at 4°C for 45 min at ~ 20,000 g and subsequently diluted ~ 3-fold in Q Buffer (20 mM Tris, pH 8.0, 10% glycerol, 5 mM BME) so that the salt concentration of resulting sample was ~ 100 mM NaCl. This sample was then loaded onto a HiTrap Q HP column that was equilibrated in Q Wash Buffer (20 mM Tris, pH 8.0, 100 mM NaCl, 10% glycerol, 5 mM BME) and washed with five column volumes of Q Wash Buffer. The xl XLF$^{\Delta KBM}$ protein was collected in the flow through which was then loaded onto a HiTrap HP SP column where majority of the sample was collected in the flow through again and subsequently concentrated using a 3-MWCO centrifugal spin filter (Amicon (MiliporeSigma), Burlington, MA, USA). The sample was then flash frozen in liquid nitrogen and stored at −80°C until use.

## xl tdXLF constructs

H10-SUMO-xl tdXLF$^{WT/WT}$, H10-SUMO-xl tdXLF$^{\Delta KBM/WT}$, H10-SUMO-xl tdXLF$^{WT/\Delta KBM}$, H10-SUMO-xl tdXLF$^{\Delta KBM/\Delta KBM}$, H10-SUMO-xl tdXLF$^{L301E/WT}$, H10-SUMO-xl tdXLF$^{WT/L301E}$, and H10-SUMO-xl tdXLF$^{L301E/L301E}$ were all expressed and purified as previously described (*Graham et al., 2018*). Each tdXLF construct was transformed into *E. coli* BL21(DE3)pLysS cells and cultures were grown at 37°C until the OD$_{600}$ was between 0.55 and 0.70. Expression was induced by adding IPTG to a 1 mM final concentration. Expression was the carried out at 22°C for 4 hr. Cultures were then spun down, washed in 1x PBS buffer, flash frozen in liquid nitrogen, and stored at −80°C. The same His10-SUMO purification steps described for wild-type XLF above were then followed for the tandem dimer constructs. The flow through from the second Ni-NTA agarose resin (Qiagen) incubation was spun at 20,000 x g for 1 hr at 4°C. The sample was then diluted with 2.5 volumes of SP Buffer (50 mM Na-MES, pH 6.5, 10% glycerol, 5 mM BME) so that final [NaCl] was 100 mM in the sample before being loaded onto a HiTrap SP HP column that was equilibrated in SP Buffer A (50 mM Na-MES, pH 6.5, 10% glycerol, 5 mM BME, 100 mM NaCl). The column was then washed with five column volumes of SP Buffer A. Protein bound to the column was eluted using a 100–1000 mM NaCl gradient over 30 column volumes. Peak fractions were pooled and concentrated using a 3- or 10 kDa MWCO centrifugal spin filter (Amicon). In the case of the H10-SUMO-tdXLF$^{\Delta KBM/\Delta KBM}$ mutant, the isoelectric point is significantly lower than that of the wild type XLF (6.13 vs. 8.00), and this mutant did not stick to the SP HP column. For this mutant, the SP HP flow through was concentrated and taken to the next step. The concentrated sample from the HiTrap SP HP column was then loaded onto a Superdex 200 Increase 10/300 GL equilibrated in (50 mM Na-MES, pH 6.5, 10% glycerol, 350 mM NaCl, 5 mM BME). Peak fractions were pooled and concentrated as described above. Samples were flash frozen in liquid nitrogen and stored at −80°C until use.

## xl XLF heterodimers

Purification of XLF heterodimers followed a previously described protocol (*Graham et al., 2018*). The Flag-Avi-xl XLF/His10-XLF and Flag-Avi-xl XLF/His10-xl XLF$^{\Delta tail \Delta KBM}$ XLF heterodimer constructs were transformed into BL21(DE3) pLysS cells along with a BirA biotin ligase expression plasmid. Cultures were grown at 37°C until the OD$_{600}$ was between 0.4 and 0.6. IPTG was then added to cultures at a 1 mM final concentration, and biotin was added at a final concentration of 25 μM to allow for BirA-dependent biotinylation of Avi-tagged proteins. Cultures were moved to 22°C for 4 hr. Cultures were then spun down to cell pellets, washed in 1x PBS buffer, flash frozen in liquid nitrogen, and stored at −80°C. Pellets were thawed and resuspended in 15 mL of His-SUMO lysis buffer (20 mM Tris-HCl, pH 8.0, 1 M NaCl, 30 mM imidazole, 5 mM BME, and 1 mM PMSF) per liter of culture and sonicated. The lysates were then spun at 20000 rpm in a SS34 fixed angle rotor for 1 hr at 4°C. For

90 min at 4°C, the supernatant was incubated with Ni-NTA agarose (Qiagen) that was equilibrated in His-SUMO lysis buffer. The Ni-NTA resin was then washed with lysis buffer. Proteins bound to the Ni-NTA resin were then eluted by incubating the Ni-NTA resin in elution buffer (20 mM Tris-HCl, pH 8.0, 350 mM NaCl, 250 mM imidazole, 5 mM BME) and incubating for 2 min. This elution step was repeated several times. The peak fractions from the Ni-NTA eluate were then pooled and passed over SoftLink Avidin resin (Promega, Madison, WI, USA). The SoftLink Avidin resin was then washed extensively with SoftLink Avidin Wash Buffer A (20 mM Tris HCl, pH 8, 1 M NaCl, 10% glycerol, 5 mM BME) to remove any proteins that do not have a biotinylated AviTag. SoftLink Avidin Wash Buffer B (20 mM Tris-HCl, pH 8, 350 mM NaCl, 10% glycerol, 5 mM BME) was then put over the resin to bring the [NaCl] down to 350 mM. Avi-tagged protein was eluted from the SoftLink Avidin resin using SoftLink Avidin Elution Buffer (20 mM Tris HCl, pH 8, 350 mM NaCl, 10% glycerol, 5 mM BME, 5 mM biotin). Peak elution fractions were pooled and concentrated using a 10 kDa MWCO centrifugal concentrator (Amicon) before being flash frozen in liquid nitrogen and stored at −80°C.

### His10-xl XLF$^{\Delta tail \Delta KBM}$

Expression and purification of His10-xl XLF$^{\Delta tail \Delta KBM}$ followed a previously described protocol (*Graham et al., 2018*). The same expression and purification steps used for the wild type His10-SUMO-xl XLF construct were used for this construct. The Ni-NTA eluate was dialyzed against MonoQ Buffer A (20 mM Tris-HCl, pH 8.0, 10% glycerol, 100 mM NaCl, 5 mM BME) at 4°C. After two rounds of dialysis, each being more than 6 hr long, the dialysate was filtered using a 0.22 μm syringe filter. The filtered sample was then loaded onto HiTrap Mono Q column equilibrated in MonoQ Buffer A. The column was washed with seven column volumes of Buffer A, and protein was eluted from the column using a 30 mL gradient of Buffer A into Buffer B (20 mM Tris-HCl, pH 8.0, 10% glycerol, 1000 mM NaCl, 5 mM BME). Peak fractions were pooled and concentrated using a 3 kDa MWCO centrifugal concentrator (Amicon). The sample was then flash frozen in liquid nitrogen and stored at −80°C until use.

## XLF subunit exchange assay

Ensemble end joining assays utilizing XLF heterodimer constructs require that the subunits do not exchange over the course of the experiment. We have previously shown that there is no subunit exchange for full length XLF constructs over a timescale of hours. We employ a similar protocol here to test whether individual XLF monomeric subunits can exchange between full length and C-terminally truncated XLF dimers (*Graham et al., 2018*). His10-XLF$^{\Delta tail \Delta KBM}$ and wt XLF were purified separately as described above. These proteins were then mixed in a 10 μL volume of protein storage buffer (20 mM Tris-HCl, pH 8, 350 mM NaCl, 10% glycerol, and 5 mM BME) so that the final concentration of each was 5 μM. This mixture was left to incubate at room temperature for 1–3 hr in a humidified chamber to prevent evaporation. After incubation the sample was spun at 16,000 rcf for 10 min at room temperature or 4°C. 100 μL of His-SUMO Lysis buffer (20 mM Tris-HCl, pH 8.0, 1 mM NaCl, 30 mM imidazole, and 5 mM BME) was added to the 10 μL mixture. A 20 μL aliquot was taken at this point and mixed with 20 μL of 2x Laemmli sample buffer (Bio-Rad, Hercules, CA, USA). A total of 80 μL of the mixture was then incubated for 45–60 min at room temperature or 4°C with 10 μL NiNTA resin (Qiagen) prewashed with His-SUMO Lysis buffer (20 mM Tris-HCl, pH 8.0, 1 M NaCl, 30 mM imidazole, 5 mM BME). This sample was then spun down, and the supernatant was collected. The NiNTA resin was then washed by resuspending the resin in 500 μL of His-SUMO Lysis buffer, spinning, removing the supernatant and mixing 20 μL with 20 μL of 2x Laemmli sample buffer, and repeating twice. 80 μL of His-SUMO Elution Buffer (20 mM Tris-HCl, pH 8.0, 350 mM NaCl, 300 mM imidazole, 5 mM BME) was then added to resuspend the NiNTA resin and incubated for 10–30 min at room temperature. Next, the sample was spun again, and the supernatant from this spin (eluate) was collected. Of the eluate, 20 μL was then mixed with 20 μL of 2x Laemmli sample buffer. The resin was then resuspended in 80 μL His-SUMO Elution Buffer and a 20 μL aliquot was taken and mixed with 20 μL of 2x Laemmli sample buffer. All samples were heated to 95°C for 5 min and cooled to room temperature.

Samples were then run on a 4–15% precast SDS-PAGE gel (Bio-Rad), transferred to polyvinylidene fluoride membranes for 16.5 hr at 30 V at 4°C, and blocked with 5% powdered nonfat milk in PBST buffer (1x phosphate-buffered saline with 0.05% Tween 20). Membranes were probed for 1 hr at

room temperature with 1:500 anti-XLF (New England Peptides, Gardner, MA, USA. Details in Materials and methods section under Immunodepletion) or 1:1000 anti-His (Bio-Rad, product code MCA1396A) in PBST with 2.5% BSA. Membranes were then washed 3x with PBST. The anti-His blot was then probed with 1:20,000 horseradish peroxidase-conjugated rabbit anti-mouse IgG (H+L) secondary antibody (Jackson ImmunoResearch) in 5% non-fat milk in PBST for 1 hr at room temperature. The anti-XLF blot was also probed for 1 hr at room temperature with 1:10,000 goat anti-rabbit IgG horseradish peroxidase-conjugated secondary antibody (Jackson ImmunoResearch, West Grove, PA U.S.A.) in 5% non-fat milk in PBST. Anti-XLF can be used to exclusively monitor wt XLF because His10-XLF$^{\Delta tail\Delta KBM}$ does not contain the C-terminal peptide used to generate the antibody (see in Methods under Immunodepletion). Membranes were then washed extensively in PBST. The anti-His membrane was incubated in substrate solution (Pierce ECL Western Blotting Substrate Kit #32106) for ~120 s. The anti-XLF membrane was incubated for ~120 s with either HyGLO chemiluminescent HRP antibody detection reagent (Denville) or substrate solution (Pierce (ThermoFisher), Waltham, MA, USA. ECL Western Blotting Substrate Kit #32106). Both membranes were imaged using an Amersham Imager 600 (GE Healthcare).

## Differential scanning fluorimetry

Differential scanning fluorimetry protein thermal shift assays were carried out using a QuantStudio 7 Flex Real-Time PCR System (Applied Biosystems, Foster City, CA, USA). Reactions were pipetted into wells in a MicroAmp FAST optical 96-well plate (Life Technologies, Carlsbad, CA, USA) and covered with MicroAmp Optical Adhesive Film (Life Technologies). Each 30 µL reaction mixture containing the XLF construct of interest at a 2.5 µM concentration and the SPYRO Orange dye at a 1x concentration (see Protein Thermal Shift Dye Kit, Applied Biosystems). Protein samples were diluted in XLF storage buffer (20 mM Tris pH 8.0, 350 mM NaCl, 5 mM BME, and 10% glycerol). After a 2 min incubation at 25°C, the temperature was raised to 99°C at 0.05°C per second. The fluorescent dye was excited and measured using 470 nm and 587 nm, respectively. Melting temperatures were determined by fitting the emission signals with the Boltzmann equation using the Protein Thermal Shift software (Life Technologies). Each replicate consisted of three 30 µL reactions for each distinct sample. The average of two replicates is plotted for each sample, with error bars representing the min and max values from the two replicates. A two-tailed, unpaired $t$ test with unequal variance and the Bonferroni correction was performed to determine if the melting temperature of the XLF mutants were significantly different from wt XLF.

## *Xenopus* egg extract preparation

Cell-free extract was prepared from eggs of *Xenopus laevis* as previously described (*Lebofsky, 2009*). The Center for Animal Resources and Comparative Medicine at Harvard Medical School (AAALAC accredited) cared for the female frogs used to produce eggs for this study. All work performed in this study were done in accordance AAALAC rules and regulations and approved by the Institutional Animal Care and Use Committee (IACUC) of Harvard Medical School.

## Immunodepletion

The XLF antibody used here is the same as previously described (*Graham et al., 2016*). This peptide antibody was generated by New England Peptide, Inc (Gardner, MA U.S.A.) using a peptide (Ac-CGASKPKKKAKGLFM-OH) corresponding to the C-terminal sequence of *Xenppus laevis* XLF. Immunodepletion of endogenous XLF within egg extract was carried out as detailed previously (*Graham et al., 2016*). Nocodozole was added to extract at 7.5 ng/µL prior to immunodepletion or prior to use in experiments if no immunodepletion was required. Unless otherwise noted, all rescue experiments here used recombinant XLF added back to extract at 75 nM (monomer concentration) to match a previous measurement of XLF concentration in *Xenopus laevis* eggs (*Wühr et al., 2014*). Mock depletions were carried out using the same protocol and IgG purified from Rabbit Serum (Gibco (ThermoFisher), Waltham, MA, USA) by protein A sepharose affinity chromatography as previously described (*Graham et al., 2016*).

## Ensemble end joining assay

The ensemble gel-based end joining time course and titration assays were performed as previously described (*Graham et al., 2016*; *Graham et al., 2018*). For each reaction condition, extract was supplemented with a 30x ATP regeneration mixture (65 mM ATP, 650 mM phosphocreatine, 160 ng/μL creatine phosphokinase) to a 1x final concentration and 25–30 ng/μL closed circular 'carrier' DNA that is required for joining of dilute linear substrates as well as for DNA replication in extract (*Graham et al., 2016*; *Lebofsky et al., 2011*). In cases where recombinant protein is added to extract and directly compared to conditions where protein was not added, the corresponding protein storage buffer was added back to those conditions without recombinant protein to ensure the volume and composition of each reaction are directly comparable. To initiate the time course end joining reactions, a radiolabeled 2.8 kb linear DNA substrate with blunt ends was added to reactions at approximately 1 ng/μL. The preparation of this substrate has been previously described (*Graham et al., 2016*; *Graham et al., 2017*). Reactions were carried out at room temperature for the indicated time. Time points were taken by removing an aliquot of the reaction mixture and stopping the reaction by addition of stop solution (80 mM Tris, pH 8.0, 8 mM EDTA, 0.13% phosphoric acid, 10% Ficoll, 5% SDS, and 0.2% bromophenol blue). The 0 min time point was taken and mixed with stop solution immediately after adding the radiolabeled substrate to the reaction and mixing. The titration end joining assay was assembled on a thermocycler at 2°C. These reactions were initiated by moving the thermocycler to 22°C for 20 min. At the 20 min time point, the thermocycler was moved back to 2°C, and stop solution was added to each reaction. All reaction samples were then digested for 1 hr at 37°C by adding proteinase K. Digested samples were then run on a Tris-borate-EDTA 0.8% agarose gel. The gel was then pressed and dried onto a HyBond-XL nylon membrane (GE Healthcare) and exposed to a storage phosphor screen. Exposed screens were scanned using a Typhoon FLA 7000 imager (GE Healthcare).

## Microscope and flow cell construction

Single-molecule experiments were performed using a through-objective TIRF microscope built around an Olympus IX-71 inverted microscope. 532 nm and 641 nm laser beams (Coherent Sapphire 532 and Coherent Cube 641) were expanded, combined using dichroic mirrors, expanded again, and focused on the rear focal plane of an oil immersion objective (Olympus UPlanSApo, 100x; NA, 1.40). The focusing lens was placed on a vertical translation stage to permit manual adjustment of the TIRF angle. The emission light was separated from the excitation light using a multipass dichroic mirror. The laser lines were further attenuated with a StopLine 488/532/635 notch filter (Semrock, Rochester, NY, USA). A home-built beamsplitter (*Graham et al., 2017*) was used to separate Cy3 and Cy5 emission signals. These two channels were imaged on separate halves of an electron-multiplying charge-coupled device camera (Hamamatsu, Hamamatsu, Japan. ImageEM 9100–13), which was operated at maximum EM gain. An automatic microstage (Mad City Labs, Madison, WI, USA) was used to position the sample and move between fields of view.

Microfluidic flow cells were constructed as previously described (*Stinson et al., 2020*; *Graham et al., 2017*). Briefly, holes for an inlet and an outlet were drilled in a glass microscope slide and PE tubing was placed into each and sealed with epoxy. A channel was cut out of a strip of double-sided SecureSeal Adhesive Sheet (Grace Bio-Labs, Bend, Oregon, USA), and this channel was placed onto the glass slide so that the two holes are at opposing ends of the channel. A glass coverslip was placed on the bottom of the flow cell on the second side of the double-sided adhesive. This coverslip was functionalized with a mixture of mPEG-SVA-5000 (Laysan Bio, Inc Arab, AL, USA) and biotin-mPEG-SVA-5000 (Laysan Bio, Inc). The edges of the channel were then sealed with epoxy.

## Single-molecule FRET SR complex formation assay

The protocol for the intramolecular end joining single-molecule FRET assay generally follows a previously established protocol (*Graham et al., 2016*; *Graham et al., 2017*). One mg/mL streptavidin in PBS buffer was flowed into the flow cell and left to incubate at room temperature for 5 min. This solution was then washed out with egg lysis buffer (ELB; 10 mM HEPES, pH 7.7, 50 mM KCl, 2.5 mM MgCl$_2$) before introducing the DNA substrate. This 2 kb blunt-ended linear DNA substrate contains an internal biotin for immobilization within the flow cell. Cy3 and Cy5 flourophores are positioned seven nucleotides from each opposing end so that energy can be transferred from Cy3 to Cy5 upon

Cy3 excitation if those opposing ends are synapsed. The construction of this substrate has been described in detail previously (*Graham et al., 2016*; *Graham et al., 2017*). The DNA substrate was flowed into the flow cell in the presence of an oxygen scavenging system (5 mM protocatechuic acid (PCA) and 100 nM protocatechuate 3,4-dioxygensae (PCD)). One mM trolox was also included and serves as a triplet state quencher. This oxygen scavenging system with trolox was included in every solution introduced into the flow cell and imaged. After ~ 5 min, the DNA solution was washed out of the flow cell with ELB supplemented with the oxygen scavenging system.

Samples of extract were immunodepleted, supplemented with the oxygen scavenging system with trolox, the ATP-regeneration system described above, and recombinant protein as described above. Recombinant protein was included at 500 nM for the experiment shown in *Figure 3* and 75 nM for the experiment shown in *Figure 4—figure supplement 3*. Data acquisition began 30 to 60 s after flowing the extract sample into the flow cell. One second exposures were collected every other second, and laser excitation was alternated so that during a single excitation cycle four 532 nm excitation frames were followed by a single 641 nm excitation frame. These excitation cycles were repeated while imaging a single field of view within the flow cell for 15 min. After 15 min, the flow cell was moved via the automatic microstage so that a new and distinct field of view was now centered under the objective and brought into focus. The same imaging procedure was repeated for 15 additional minutes in this new field of view. This process was repeated so that distinct fields of view were imaged from 0 to 15 min, 15–30 min, and 30–45 min. Previously described custom MATLAB scripts were used to analyze the data and are provided here as (*Source code 1* and *Source code 2*; *Graham et al., 2016*; *Graham et al., 2018*; *Graham et al., 2017*). Trajectories were truncated prior to photobleaching events that were detected automatically during data processing and analysis. Additionally, trajectories were excluded if they exhibited low signal-to-noise, multistep photobleaching, large fluctuations in fluorescence intensity not due to FRET, or if there was more than one peak present within a region of interest. SR complex formation events were identified manually using five consecutive frames above a FRET threshold (0.25) as a guideline. The rate of SR complex formation was calculated by dividing the number of SR complex formation events by the total time that SR complex formation was possible (the total time that Cy3 and Cy5 emission signal are both present and the ends were not already joined, summed over all substrate molecules that were tracked for a certain experimental condition). Sample sizes are reported in *Supplementary file 1*. The plot in *Figure 3G* was generated using the notBoxPlot MATLAB function (*Campbell, 2020*). Single-molecule FRET histograms were generated using a bin number equal to the square root of N, where N is the number of data points from the smallest dataset out of the nine datasets shown in *Figure 3D–F*. Those bins span the same range of values for all datasets shown so that any bin is directly comparable between datasets.

## DNA pulldown assay

The DNA pulldown assay was performed essentially as previously described (*Stinson et al., 2020*). This protocol is outlined below with any alterations described in detail.

Biotinylated primers were used to generate a 1 kb DNA fragment with biotin molecules attached to the 5′ termini. Streptavidin-coated magnetic beads (36 µL per biological replicate) were washed twice in 2x Bead Wash Buffer (10 mM Tris, pH 7.4, 2 M NaCl, and 20 mM EDTA), and subsequently resuspended in 1x Bead Wash Buffer with 30 nM of the biotinylated DNA substrate described above at room temperature for 20 min. The DNA-bound beads were again washed twice in 2x Bead Wash Buffer and then washed twice in 1x Cutsmart Buffer (New England Biolabs (NEB), Ipswich, MA, USA) before being resuspended in 80 µL of 1x Cutsmart Buffer. The DNA-bead suspension was then split so that plasmid control and DSB samples could be prepared and tested in parallel. One µL of Xmn1 (NEB) was added to the DSB sample to generate the DSB, and both samples were incubated at 37°C for 6 hr. Samples were then washed 3x with 2x Bead Wash Buffer and once in Egg Lysis Blocking Buffer (10 mM HEPES, pH 7.7, 50 mM KCl, 2.5 mM MgCl$_2$, 250 mM sucrose, and 0.02% Tween20) to remove the Xmn1 and reduce non-specific binding to the beads. Beads were then incubated in Egg Lysis Blocking Buffer for 20 min on ice and then resuspended in 50 µL of Egg Lysis Blocking Buffer.

Extract was supplemented with the 30x ATP regeneration mixture (65 mM ATP, 650 mM phosphocreatine, 160 ng/µL creatine phosphokinase) to a 1x final concentration and with 30 ng/µL of circular pBlueScript II plasmid to act as carrier DNA (*Lebofsky et al., 2011*). Of the DNA-beads sample, 15 µL was mixed with an equal volume of extract and incubated for 15 min at room

temperature. Thirty µL of the reaction was then layered over 180 µL of ELB-Sucrose Cushion Solution (10 mM HEPES, pH 7.7, 50 mM KCl, 2.5 mM MgCl$_2$, 500 mM sucrose) in a horizontal rotor (Kompspin, Ku Prima-18R). The pelleted DNA-beads were then washed in 180 µL of Egg Lysis Blocking Buffer and resuspended in 20 µL of 1x reducing Laemmli sample buffer. Extract was diluted 1:40 in 1x reducing Laemmli sample buffer to be used as input sample for western blotting. Samples were separated on a 4–15% precast SDS-PAGE gel (BioRad) for 30 min at 200V and subsequently transferred to a PVDF membrane for 60 min at 103V at 4°C. Membranes were blocked with 5% powdered nonfat milk dissolved in 1x PBST for 30 min and then incubated with primary antibody diluted in 1xPBST containing 2.5% BSA (OmniPur, (MiliporeSigma), Burlington, MA, USA) for 12–16 hr at 4°C. Primary antibodies were used at the following concentrations: ∝-Ku80 1:10,000, ∝-Lig4 1:2000, ∝-XRCC4 1:2000, ∝-XLF 1:500, ∝-DNA-PKcs 1:5000, and ∝-Orc2 1:10,000. After extensive washing with 1x PBST, membranes were incubated with goat anti-rabbit-HRP (Jackson ImmunoResearch) secondary antibody diluted 1:10,000 or 1:20,000 or rabbit anti-mouse-HRP (Jackson ImmunoResearch) secondary antibody diluted 1:10,000 in 5% powered nonfat milk and 1x PBST for 1 hr at room temperature. Membranes were washed again with 1x PBST, incubated with HyGLO Quick Spray (Denville Scientific, Metuchen, NJ U.S.A.), and imaged on an Amersham Imager 600 (GE Healthcare).

## Cellular GFP reporter NHEJ assay

Xlf-/-mESCs with the EJ7-GFP reporter integrated at the Pim1 locus, Cas9/sgRNA plasmids for targeting the DSBs to this reporter (i.e. px330-7a and px330-7b), and pCAGGS-BSKX empty vector (EV), were previously described (*Bhargava et al., 2018*). The Xlf-/-EJ7 GFP reporter mESC line was generated in a prior study that included validation by XLF immunoblotting and negative mycoplama testing (*Bhargava et al., 2018*). Cas9/sgRNA plasmids used the px330 plasmid (Addgene, Watertown, MA, USA product # 42230).

For the reporter assays, cells were seeded on a 24-well plate, and subsequently transfected with 200 ng of px330-7a, 200 ng px330-7b, and 50 ng of EV or XLF expression vector, using 1.8 µL of Lipofectamine 2000 (ThermoFisher) in 0.6 mL total volume. Three days after transfection, cells were analyzed by flow cytometry using a CyAN ADP (Dako (Agilent), Santa Clara, CA, USA), as described in *Bhargava et al., 2018*. GFP+ frequencies were normalized to parallel transfections with a GFP+ expression vector (pCAGGS-NZE-GFP) (*Bhargava et al., 2018*).

## Immunobloting analysis

For immunoblotting analysis, cells were transfected as for the reporter assays, but scaled twofold onto a 12 well, and replacing the Cas9/sgRNA plasmids with EV. Subsequently, cells were extracted using NETN buffer (20 mM Tris at pH 8.0, 100 mM NaCl, 1 mM EDTA, 0.5% Igepal, 1.25 mM DTT, Roche (Basel, Switzerland) protease inhibitor) with several freeze/thaw cycles. Extracts were probed with antibodies for mouse monoclonal anti-FLAG HRP (Sigma-Aldrich Cat#A8592), or rabbit polyclonal anti-ACTIN (Sigma-Aldrich (MiliporeSigma) Cat#A2066) with the secondary antibody goat polyclonal Anti-Rabbit IgG HRP (Abcam, Cambridge, U.K. Cat#ab205718). ECL western blotting substrate (ThermoFisher Cat#32106) was used to develop HRP signals.

## Size exclusion chromatography with multi-angle light scattering

SEC-MALS experiments were performed on an Agilent 1260 Infinity Isocratic Liquid Chromatography System coupled to a Wyatt (Santa Barbara, CA, USA) Dawn Heleos II Multi-Angle Light Scattering (MALS) detector and a Wyatt Optilab T-rex Refractive Index Detector. A Sepax SRT SEC-300 column was equilibrated in SEC buffer (50 mM Na-MES, pH 6.5, 350 mM Tris, 10% glycerol, and 1 mM DTT) at 0.1 mL/min flow rate and subsequently at 0.5 mL/min flow rate for 1 hr. As a control, 100 µL of 2 mg/mL BSA solution (ThermoFisher (Pierce) Cat#23209) was run over the column, which produced expected monomer and dimer peaks. Prior to running XLF samples through SEC-MALS, they were put through a Superdex 200 Increase 10/300 GL column in the same SEC buffer. For each sample, a single, strong peak was observed eluting from the Superdex 200 increase 10/300 GL column. Peak fractions were pooled and concentrated using 0.5 mL Amicon centrifugal concentrator with a 3 kDa MW cutoff. For each XLF sample, 100 µL of sample (10–15 µg) was injected into the SEC-MALS system for analysis. The ASTRA software suite (Wyatt) was used to process and analyze all SEC-MALS data.

## Acknowledgements

We thank TGW Graham for generating and sharing the His10-SUMO-xl XLF, His10-SUMO-xl XLF[L117D], His10-SUMO-xl XLF[WT/WT], and His10-xl XLF: Flag-Avi-xl XLF plasmids and for his MATLAB smFRET analysis scripts, K Arnett (HMS Center for Macromolecular Interactions) for assistance with SEC-MALS and DSF assays, and members of the Loparo lab, in particular BM Stinson, for constructive feedback. We thank Johannes Walter and his laboratory for access to their frog facility, for sharing reagents, and for helpful discussions.

## Additional information

### Funding

| Funder | Grant reference number | Author |
|---|---|---|
| National Institutes of Health | R01GM115487 | Joseph J Loparo |
| National Institutes of Health | R01CA197506 | Jeremy M Stark |
| National Institutes of Health | R01CA240392 | Jeremy M Stark |
| National Institutes of Health | F32GM129913 | Sean M Carney |
| National Institutes of Health | T32 GM008313 | Sadie C Piatt |

The funders had no role in study design, data collection and interpretation, or the decision to submit the work for publication.

### Author contributions

Sean M Carney, Conceptualization, Resources, Formal analysis, Funding acquisition, Validation, Investigation, Methodology, Writing - original draft, Writing - review and editing; Andrew T Moreno, Investigation, Methodology, Writing - review and editing; Sadie C Piatt, Resources, Validation, Investigation, Writing - review and editing; Metztli Cisneros-Aguirre, Formal analysis, Validation, Investigation, Writing - review and editing; Felicia Wednesday Lopezcolorado, Validation, Investigation, Writing - review and editing; Jeremy M Stark, Conceptualization, Formal analysis, Supervision, Investigation, Project administration, Writing - review and editing; Joseph J Loparo, Conceptualization, Supervision, Funding acquisition, Writing - original draft, Project administration, Writing - review and editing

### Author ORCIDs

Sean M Carney https://orcid.org/0000-0002-2674-1064
Felicia Wednesday Lopezcolorado http://orcid.org/0000-0002-2916-6042
Jeremy M Stark https://orcid.org/0000-0002-2625-5373
Joseph J Loparo https://orcid.org/0000-0003-4941-4696

### Ethics

Animal experimentation: The Institutional Animal Care and Use Committee (IACUC) of Harvard Medical School approved of all work performed in this study (Protocol# IS00000051-6), which was done in accordance with AAALAC rules and regulations.

### Decision letter and Author response

Decision letter https://doi.org/10.7554/eLife.61920.sa1
Author response https://doi.org/10.7554/eLife.61920.sa2

## Additional files

### Supplementary files

- Source code 1. Peak Finder Script.
- Source code 2. Trajectory Analysis Script.

- Supplementary file 1. Single-molecule FRET data table. Table containing replicate number, number of molecules tracked, number of SR complex formation events detected, average SR complex formation rate, and the standard error of the mean for the rate for each experimental condition.

- Supplementary file 2. Dataset replicate and statistics table. Table containing figure panel reference, type of data, replicate number, data shown, and reference to details of any statistical analysis for each data-based figure panel.

- Transparent reporting form

### Data availability

Source data files for all summary graphs have been provided. The MATLAB scripts used to analyze and generate the results shown in Figure 3 and Figure 4—figure supplement 3 are also included.

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
