## [Decision Letter]

**Acceptance summary:**

Synapse of the broken DNA ends is a critical step in DNA double-strand break repair by non-homologous end joining (NHEJ). XRCC4-like factor (XLF) is an important player in this process. This manuscript builds on the lab's prior work, which suggested a model whereby a single XLF dimer is able to sustain NHEJ (Graham et al., 2018). Current work presents a careful study that investigates the role of the unstructured tail of XLF. Using ensemble reactions in *Xenopus* extracts, single-molecule experiments, and functional cell-based assays, the authors convincingly show that only one tail of the XLF dimer, of sufficient length and ending with the Ku-binding motif (KBM), is needed to sustain NHEJ. Thus a single XLF-Ku contact is sufficient for the two ends to join.

**Decision letter after peer review:**

Thank you for submitting your article "XLF acts as a flexible connector during non-homologous end joining" for consideration by *eLife*. Your article has been reviewed by three peer reviewers, and the evaluation has been overseen by Maria Spies as a Reviewing Editor and Cynthia Wolberger as the Senior Editor. The following individual involved in review of your submission has agreed to reveal their identity: Terence Strick (Reviewer #3).

The reviewers have discussed the reviews with one another and the Reviewing Editor has drafted this decision to help you prepare a revised submission.

Summary:

Synapsis of the broken DNA ends is a critical step in DNA double-strand break repair by non-homologous end joining (NHEJ). XRCC4-like factor (XLF) is an important player in this process. This manuscript builds on the lab's prior work, which suggested a model whereby a single XLF dimer is able to sustain NHEJ (Graham et al., 2018). Current work presents a careful study that investigates the role of the unstructured tail of XLF. Using ensemble reactions in *Xenopus* extracts, single-molecule experiments, and functional cell-based assays, the authors convincingly show that only one tail of the XLF dimer, of sufficient length (but not its sequence) and ending with the Ku-binding motif (KBM), is needed to sustain NHEJ. Thus a single XLF-Ku contact is sufficient for the two ends to join.

After discussing the paper, the reviewers agree that the data are solid and substantiate the key points of the authors' findings; the paper is well written and concise. The main point of the reviewers' discussion concerned whether the authors' observation that a single XLF-Ku contact is sufficient for the two DNA ends to join unambiguously excludes the role of XLF-XRCC4 filaments in NHEJ. The authors state that they disfavor the filament model. In light of the reviewers' argument, which did not yield a clear agreement, a more thorough discussion of the XLF dimer vs. XLF/XRCC4 filaments will help the readers to navigate the current models of NHEJ.

Essential revisions:

1) Please expand the discussion of the XLF dimer vs. XLF/XRCC4 filaments models. While a single XLF-Ku contact seems sufficient (as clearly shown here) for the two DNA ends to join when the DBS is ready for ligation, the filaments may play a role earlier in the pathway before the actual repair process that is mediated by KU and ligation (e.g. as proposed by Brouwer et al., 2016). It will be informative for the readers and the field to delineate the time and place for the proposed step-by-step mechanism and/or provide a more articulated argument against the filament.

2) A tandem affinity purification approach is used to isolate XLF heterodimers. Both dimers and higher order oligomers can elute from both affinity resins. A calibrated SEC trace could be informative to distinguish the species.

3) Do the heterodimer mutants promote the Short Range (SR) complex in the FRET assay?

---

## [Author Response]

Essential revisions:1) Please expand the discussion of the XLF dimer vs. XLF/XRCC4 filaments models. While a single XLF-Ku contact seems sufficient (as clearly shown here) for the two DNA ends to join when the DBS is ready for ligation, the filaments may play a role earlier in the pathway before the actual repair process that is mediated by KU and ligation (e.g. as proposed by Brouwer et al., 2016). It will be informative for the readers and the field to delineate the time and place for the proposed step-by-step mechanism and/or provide a more articulated argument against the filament.

We appreciate the reviewer’s concern and have now substantially added to our discussion of XLF-XRCC4 filaments and their potential role in NHEJ. Please see the revised text below.

“How does the tail of XLF contribute to NHEJ? […] During repair, Lig4, which is required to recruit and stabilize XRCC4 at DNA ends (Costantini et al., 2007; Wu et al., 2007), likely blocks XLF molecules from interacting with both XRCC4 monomers thereby preventing filament formation (Recuero-Checa et al., 2009; Ochi et al., 2012).

2) A tandem affinity purification approach is used to isolate XLF heterodimers. Both dimers and higher order oligomers can elute from both affinity resins. A calibrated SEC trace could be informative to distinguish the species.

We have now performed SEC-MALS experiments on both heterodimer constructs, Flag-Avi-XLF/H10-XLF and Flag-Avi-XLF/H10-XLF^ΔTailΔKBM^, as well as wild type XLF. This data is shown in a new figure, Figure 4—figure supplement 2D, and figure legends and Materials and methods sections have been updated to reflect these new data. Each sample produced a single peak that corresponded to a molecular mass that measured within 10% of the expected molecular weight of a dimer for each construct. The XLF tandem dimer has previously been shown to exist solely as a dimer (Graham, et al., 2018). We believe these data provide strong evidence that the XLF heterodimer constructs used in this manuscript exist primarily as dimers.

3) Do the heterodimer mutants promote the Short Range (SR) complex in the FRET assay?

We have now done single-molecule FRET experiments to show that the heterodimer mutant, Flag-Avi-XLF /H10-XLF^ΔtailΔKBM^ forms the SR complex, although less efficiently than the Flag-Avi-XLF /H10-XLF heterodimer. This defect is likely due to reduced recruitment as the mutant only has a single KBM instead of two. However, we cannot formally rule out contributions from other unknown mechanisms involving two C-terminal tails that could enhance end joining. We have updated the text (see below) to incorporate these results. The data from these new experiments are shown in Figure 4—figure supplement 3A, and example trajectories are shown in Figure 4—figure supplement 3B-D. Supplementary file 1 and Supplementary file 2 have been updated to include the data from these experiments, and the source data for Figure 4—figure supplement 3A are available. We have modified the text in the Results section to address these new data (see below).

“Size-exclusion chromatography and multi-angle light scattering (SEC-MALS) experiments confirmed that these heterodimer constructs exist as dimers in solution (Figure 4—figure supplement 2D). […] Overall, these results demonstrate that a single KBM on a single tail of XLF is sufficient for robust end joining.”